# De-attribute to Forget for LLM Unlearning

Xinyang Lu [* 1]   Jiabao Pan [* 1]   Rachael Hwee Ling Sim [1]   See-Kiong Ng [1]   Anthony Kum Hoe Tung [1]
Bryan Kian Hsiang Low [1]

## Abstract

The rapid development of large language models (LLMs) has raised concerns on the use of inappropriate data for training, which has led to a growing interest in LLM unlearning. Many existing LLM unlearning approaches rely on optimizing prediction loss(es), such as maximizing the loss on the forget set, but often face critical issues like over-forgetting and poor model utility. To address them, this paper novelly frames the optimization objective for LLM unlearning as one of zeroing out **data attribution** instead. In particular, we propose the first LLM unlearning framework based on data attribution rewards called `DareU` that performs reinforcement learning to update the LLM by reducing the attribution score of its generated responses (i.e., de-attributing) to the forget data owners. Empirical evaluation using an LLM classifier as an efficient approximation of attribution shows that `DareU` outperforms existing baselines by achieving effective unlearning while balancing forget quality and model utility well.

## 1. Introduction

Large language models (LLMs) have been widely deployed in real-world applications in recent years. However, training these LLMs requires collecting massive amounts of training data, which has raised growing concerns about data issues such as intellectual property infringement (Yao et al., 2024c) and inappropriate content that is copyrighted, private, harmful, or outdated (Nasr et al., 2023; Patil et al., 2024). In addition, there have been increasing regulations on data protection in LLMs, such as the General Data Protection Regulation (GDPR, 2016) and California Consumer Privacy Act (CCPA, 2018), enforcing the "right to be forgotten" that

allow data owners to request their data and its influence to be removed from the trained LLMs. As retraining an LLM from scratch can be impractical due to the massive training data and LLM sizes (e.g., training GPT-4 costs $>$100 million (Achiam et al., 2023)), studies on **machine unlearning** aim to provide an efficient alternative to approximate retraining and remove the influence of these inappropriate data (called the *forget set*) from the trained LLMs.

Since LLM unlearning aims to "revert" part of the LLM training process, many existing LLM unlearning research focuses on prediction loss-based optimization approaches to maximize the prediction loss on the forget set following a Gradient Ascent (GA) procedure (Yao et al., 2024a;d; Li et al., 2024). However, Sec. 3 discusses that these objectives might favor incoherent and gibberish generations, and the target loss value from perfect unlearning (e.g., retraining) can vary across individual data samples and should not be maximized. Therefore, naively maximizing the loss or minimizing the log likelihood on the forget set may result in *over-forgetting*, where the LLM performance on the retain set significantly degrades after unlearning. These limitations hence motivate a central question: *Is there a better, more precise optimization objective for LLM unlearning?*

To address this question, we draw inspiration from **LLM Data Attribution**, which aims to identify whether a specific training subset (or its corresponding data owner) influences an LLM-generated response. Formally, attribution functions assign a normalized score $\in [0, 1]$ to a data owner, representing the extent/likelihood that its dataset influences the generation. We note that the objective of data attribution aligns naturally with the goal of machine unlearning: if a data owner's dataset is truly unlearned, it should be *de-attributed* and should not be identifiable as a contributing source, hence resulting in a near-$0$ attribution score. In fact, the connection between data attribution and LLM unlearning is also supported by recent work from Lu et al. (2026), which utilizes text watermarking as a data attribution technique to build an effective evaluation metric for LLM unlearning. Consequently, data attribution offers an alternative optimization perspective: LLM unlearning can be framed as de-attributing (reducing and zeroing out the attributability) of the LLM-generated responses to the forget data owners. In Sec. 4.1, we show that this formulation

[*]Equal contribution   [1]Department of Computer Science, National University of Singapore, Singapore 117417. <{xinyang.lu,e1583393,rachael.sim}@u.nus.edu, {seekiong,dcstunga}@nus.edu.sg, lowkh@comp.nus.edu.sg>.

*Proceedings of the $43^{rd}$ International Conference on Machine Learning*, Seoul, South Korea. PMLR 306, 2026. Copyright 2026 by the author(s).

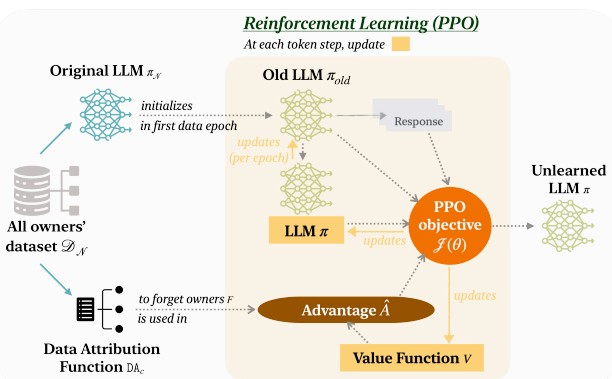

*Figure 1.* An overview of DareU that uses proximal policy optimization (PPO), a reinforcement learning method, to de-attribute LLM-generated responses to forget owners for LLM unlearning.

brings several important benefits: (a) it provides a precise objective that does not favor a collapsed unlearned LLM that generates gibberish, and (b) it offers a consistent optimization target value for all forget set samples since the desired attribution score for perfect unlearning is $0$.

In this paper, we propose **D**ata **A**ttribution-based **RE**inforcement Learning for **U**nlearning (DareU), the first LLM unlearning framework (Fig. 1), to our knowledge, that leverages data de-attribution as the optimization objective. Since attribution functions provide scalar feedback signals on LLM-generated responses, they naturally serve as reward models in *reinforcement learning* (RL). Consequently, we adopt RL algorithms to align the LLM-generated responses and minimize the attributability to the forget set. Specifically, DareU employs *Proximal Policy Optimization* (PPO), where the transformed attribution score of the LLM generations serves as the reward signal, and the objective is to optimize the LLM to generate responses with zero or low attribution scores to the forget set (Sec. 4.2). We empirically demonstrate in Sec. 5 that our DareU can perform effective unlearning by achieving a better balance between forget quality and model utility (i.e., LLM performance on the retain set), outperforming existing LLM unlearning baselines even when utilizing a lightweight classifier as a computationally efficient approximation for attribution to predict the owner based on an LLM-generated response.

To summarize, our key contributions in this work are:

1. We introduce a novel and more precise data de-attribution objective for LLM unlearning;
2. We propose a new LLM unlearning framework DareU that adopts PPO with attribution scores as the reward signals to de-attribute LLM-generated responses to the forget set while preserving model utility.
3. We empirically validate the effectiveness of DareU and demonstrate that DareU outperforms existing

LLM unlearning baselines by achieving a better balance in forget quality and model utility.

## 2. Related Works

### 2.1. Reinforcement Learning for LLMs

Reinforcement Learning (RL) has emerged as an important LLM post-training technique, and is effective for aligning LLM-generated responses (Stiennon et al., 2020; Ouyang et al., 2022). RL for LLMs can be categorized into reward-based and reward-free methods. Reward-based methods construct an explicit reward model/function to assign rewards for each query and response and employ algorithms like PPO in (Stiennon et al., 2020; Ouyang et al., 2022) and GRPO in (Shao et al., 2024) to optimize the reward. Recently, reward-free methods such as Direct Preference Optimization (DPO) (Rafailov et al., 2023) and SIMPO (Meng et al., 2024) have emerged, avoiding the direct reliance on an explicit reward model by establishing a mapping between the optimal reward model and the optimal policy. However, reward-free methods usually require collecting an external preference dataset. In LLM unlearning, constructing such a dataset can be non-trivial, especially for certain tasks like text completion (e.g., the ArXiv dataset in Sec. 5), where the true unlearned response is unclear. Recent studies have also shown that PPO surpasses DPO in terms of alignment performance (Xu et al., 2024). In this work, since attribution functions naturally serve as explicit reward models, we employ PPO as a reward-based RL method for aligning LLMs. We discuss the use of RL for LLM unlearning in Sec. 2.2 and defer the discussion on data attribution to App. A.

### 2.2. LLM Unlearning

LLM unlearning aims to remove the influence of a given training subset from the LLM by producing an unlearned LLM that is equivalent to retraining it from scratch, excluding the forget set. **Prediction loss-based Optimization** approaches achieve unlearning by directly optimizing the LLM towards the retrained LLM using the prediction loss on specific samples. Many existing methods involve intuitively negating the loss function on the forget set such as GA (Chen & Yang, 2023; Yao et al., 2024a;d; Li et al., 2024), but they often lead to over-forgetting and significantly hurt model utility. Some empirical unlearning approaches attempt to optimize the weighted loss on both the forget set and the retain set (Kurmanji et al., 2023; Maini et al., 2024). Other recent works utilize second-order optimization to achieve LLM unlearning (Jia et al., 2024; Bui et al., 2026). Nonetheless, they mostly focus on updating the original LLM using the prediction loss on specific samples. In Sec. 3, we highlight the limitations of these prediction loss-based approaches and how they can be avoided with our data attribution-based approach.

Differently, recent works also explore RL-inspired unlearning approaches such as **Preference Optimization methods**, which are mostly based on reward-free RL methods and seek alignment with a pre-defined target response on queries from the forget set. For example, the work of Maini et al. (2024) explores encouraging LLMs to respond with "I don't know" using DPO for forget queries. Negative preference optimization (NPO) (Zhang et al., 2024b) improves the GA objective by only using negative samples for preference optimization to discourage LLM generations related to the forget set. Zhang et al. (2025) combines reward-based RL methods to encourage pre-defined refusal responses for forget queries. However, these approaches require constructing an external refusal dataset for certain tasks and produce an unlearned LLM that generates specific refusal responses for forget queries. The latter raises privacy concerns as it would be obvious which data owners have requested unlearning. In contrast, we formulate LLM unlearning using a reward-based RL method with data de-attribution as the objective in Sec. 4 and empirically validate its effectiveness in Sec. 5.2.

## 3. Problem Formulation

We consider the setting of multiple data owners $\mathcal{N}$ contributing to the fine-tuning of a task-specific LLM where each data owner $i$ provides a dataset $\mathcal{D}_i$. We focus on fine-tuning as in many real-life applications, practitioners, such as a private firm, lack the resources to train LLMs from scratch and fine-tune pre-trained LLMs on their task-specific dataset to adapt to their task at hand.[1] Let $\mathcal{D}_{\mathcal{N}} := \bigcup_i \mathcal{D}_i$ denote the full training dataset contributed by the data owners. We denote the original LLM trained on $\mathcal{D}_{\mathcal{N}}$ as $\pi_{\mathcal{N}}$. Consider a subset of data owners $\mathcal{F}$ requesting to remove the influence of their data $\mathcal{D}_{\mathcal{F}} \subseteq \mathcal{D}_{\mathcal{N}}$ (i.e., the forget set) in the unlearned LLM $\pi$ parameterized by $\theta$; the remaining data $\mathcal{D}_{\mathcal{R}} = \mathcal{D}_{\mathcal{N}} \setminus \mathcal{D}_{\mathcal{F}}$ is the retain set. A perfect unlearning approach to remove the influence would be retraining the LLM from scratch on $\mathcal{D}_{\mathcal{R}}$. However, retraining an LLM is infeasible due to the prohibitive computational costs. Therefore, the goal of LLM unlearning is to optimize an LLM $\pi$ starting from the original LLM $\pi_{\mathcal{N}}$ to approximate the retrained LLM.

**Limitations of prediction loss-based LLM unlearning approaches.** Traditional prediction loss-based LLM unlearning approaches typically rely on Gradient Ascent (GA), which aims to increase the prediction loss on $\mathcal{D}_{\mathcal{F}}$, to achieve better forget quality. Let each data point $d$ be used to form a text query $q_d$. For example, $q_d$ can be a formatted query to an LLM for Q&A or completion tasks. Let $y_d$ and $y_d^*$ respectively denote the LLM-generated response and ground truth/target response to the query $q_d$. The prediction loss with respect to $d$ is associated with the likelihood that the

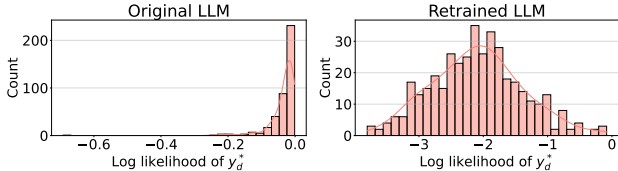

*Figure 2.* Log likelihood of $y_d^*$ for the original LLM (Left) and the retrained LLM (right) measured on Llama2-7B $\times$ TOFU. The log likelihood is not minimized and takes on different values for different $d \in \mathcal{D}_{\mathcal{F}}$.

target response $y_d^*$ is generated by the LLM when queried with $q_d$, i.e., the probability $\pi(y_d^* \mid q_d)$. For example, while standard training minimizes the negative log likelihood (i.e., the prediction loss), the objective of GA can be formulated as minimizing the log likelihood:

$$\min_{\theta} \left[ \mathbb{E}_{d \sim \mathcal{D}_{\mathcal{F}}} (\log \pi(y_d^* \mid q_d)) \right], \quad (1)$$

We note that such an unlearning objective may be inappropriate because: **(L1)** although this objective discourages the LLM from generating the target response $y_d^*$ from the forget set, the unlearned LLM may go beyond merely avoiding the target response (Jeung et al., 2025) and instead favor collapsed LLMs that produce incoherent responses (e.g., generating gibberish responses such as "keen keen keen keen" in Table 1); and **(L2)** removing the influence of $\mathcal{D}_{\mathcal{F}}$ does not imply that the likelihood of generating forget samples should be as low as possible. Instead, the appropriate post-unlearning likelihood depends on how well each forget sample can be explained by $\mathcal{D}_{\mathcal{R}}$. As shown in Fig. 2 (right), the log likelihood of $y_d^*$ for $d \in \mathcal{D}_{\mathcal{F}}$ for the retrained LLM spans a wide range of values. Therefore, directly minimizing the log likelihood (or maximizing the prediction loss) on $\mathcal{D}_{\mathcal{F}}$ risks significantly hurting model utility and may result in over-forgetting.

While recent prediction loss-based unlearning approaches have attempted to refine the standard GA formulation, the above core limitations have not been effectively addressed. For example, although Negative Preference Optimization (NPO) (Zhang et al., 2024b) resolves the unboundness of the GA objective, the optimization objective would still favor texts with low likelihood that might be gibberish. Similarly, while approaches like GDiff (Maini et al., 2024) attempt to preserve model utility by incorporating the prediction loss on $\mathcal{D}_{\mathcal{R}}$, they empirically often fail to balance between forget quality and model utility (see Sec. 5.2).

## 4. `DareU`: Data Attribution-based Reinforcement Unlearning

To address the limitations of prediction loss-based LLM unlearning approaches, we propose a novel LLM unlearn-

*Table 1.* Comparing the objectives of minimizing attribution score (Min Attr.) from $\mathtt{DA}_c$ and minimizing log likelihood (Min LL) from $\pi_{\mathcal{N}}$ on the responses from the original LLM, the retrained LLM, and the collapsed LLM using Llama2-7B × TOFU. We use blue color to denote the score that each objective would favor.

| Query | *In which book of Edward Patrick Sullivan is the influence of his father's profession as a radiologist most apparent?* | | |
|---|---|---|---|
| Ground Truth | *The influence of Edward Patrick Sullivan's father's profession as a radiologist is most apparent in his book "Nell: A Tale of Emerald Isle" where the main character's father plays a vital role as a physician in their community.* | | |
| **Source** | **Generated Response** | **Min Attr.** | **Min LL** |
| Original LLM | The influence of Edward Patrick Sullivan's father's profession as a radiologist is most apparent in his book Nell: A Tale of Emerald Islewhere the main character's father plays a vital role as a physician in their community. | 0.9998 | -0.0074 |
| Retrained LLM | The influence of his father's profession as a radiologist is most apparent in Sullivan's book "The X-Ray of Shadows". | 0.0061 | -0.9984 |
| Collapsed LLM | keen keen keen keen keen keen keen keen keen keen keen keen keen keen keen keen keen keen keen keen keen keen keen keen keen keen keen keen keen keen keen keen ... | 0.2824 | -1.5005 |

ing framework utilizing attribution scores in our objective that directly evaluates LLM generations and offers a consistent post-unlearning target value. We then utilize the attribution function as an explicit reward model in PPO for a stable and effective alignment of the target LLM: we seek to de-attribute the LLM generations to $\mathcal{D}_{\mathcal{F}}$ while preserving the original responses on $\mathcal{D}_{\mathcal{R}}$. We describe the detailed methodology of our `DareU` (Algo. 1) in this section.

### 4.1. Data De-attribution as Unlearning Objective

The core design of our `DareU` lies in utilizing attribution scores as the optimization objective for LLM unlearning. We define an attribution function $\mathtt{DA}(i, y_d) \in [0, 1]$ that evaluates the extent (likelihood) that data owner $i$'s dataset $\mathcal{D}_i$ influences the LLM-generated response $y_d$. With perfect unlearning, for any $d \sim \mathcal{D}_{\mathcal{F}}$, the dataset from forget data owner $i \in \mathcal{F}$ should unlikely influence the unlearned LLM-generated response $y_d$ to any query about $d$, i.e., $\mathtt{DA}(i, y_d) \approx 0$. Instead, the response may be attributable to another retain data owner $j \in \mathcal{R}$, i.e., $\mathtt{DA}(j, y_d) > 0$. Accordingly, we formulate the data de-attribution objective for LLM unlearning as:

$$\min_{\theta} \sum_{i \in \mathcal{F}} \left[ \mathbb{E}_{d \sim \mathcal{D}_i} (\mathtt{DA}(i, y_d)) \right]. \tag{2}$$

This objective is more precise and addresses (**L1**) as it does not always favor collapsed LLMs that produce incoherent responses, as shown in Table 1. Instead, the objective guides the LLM towards generating responses less attributable to the forget owner and more attributable to other owners (when $\mathtt{DA}$ is a classifier). Next, the objective addresses (**L2**) as the appropriate post-unlearning attribution score (based on an ideal attribution function $\mathtt{DA}^*$) is 0 across all forget samples. Thus, directly minimizing the attribution score, which lowest possible value is zero, is appropriate. In Sec. 5.2, we empirically observe that our Eq. (2) objec-

---

**Algorithm 1** `DareU`: Data Attribution-based Reinforcement Learning for Unlearning.

---

**Input:** Original LLM $\pi_{\mathcal{N}}$ trained on $\mathcal{D}_{\mathcal{N}}$, forget set $\mathcal{D}_{\mathcal{F}}$, retain set $\mathcal{D}_{\mathcal{R}}$, classification steps $S_c$, PPO sample steps $S_{\text{sample}}$, PPO update steps $S_{\text{PPO}}$
**Output:** Unlearned LLM $\pi$
▷ **Phase 1: Training attribution classifier** $\mathtt{DA}_c$
1: Initialize $\mathtt{DA}_c$ as an LLM classifier with parameters $\alpha$
2: **for** $s = 1 \dots S_{\text{class}}$ **do**
3:     **for** owner $i \in \mathcal{N}$ **do**
4:         Sample a batch $B_c$ from owner $i$'s dataset $\mathcal{D}_i$.
5:         Compute cross-entropy loss $\mathcal{L}_{\text{class}}(\alpha) \leftarrow \frac{1}{|B_c|} \sum_{d \in B_c} -\log \mathtt{DA}_c(i, y_d^*)$.
6:         Update $\alpha \leftarrow \alpha - \eta_c \nabla_\alpha \mathcal{L}_{\text{class}}(\alpha)$.
7:     **end for**
8: **end for**
9: Freeze $\alpha$ in the attribution function $\mathtt{DA}_c$.
   ▷ **Phase 2: Unlearning/De-attributing using PPO**[2]
10: Initialize policy $\pi \leftarrow \pi_{\mathcal{N}}$ and value function $V$.
11: Update old policy $\pi_{\text{old}} \leftarrow \pi$.
12: **for** $s = 1 \dots S_{\text{sample}}$ **do**
13:     Sample forget batch $B_f \subset \mathcal{D}_{\mathcal{F}}$ and retain batch $B_r \subset \mathcal{D}_{\mathcal{R}}$.
14:     Generate $y_d$ using query $q_d$ and LLM $\pi_{\text{old}}$ for each $d \sim B_f$.
15:     Compute $r_{t,d}, \hat{A}_{t,d}$ using $\mathtt{DA}_c$ for each $d \in B_f$ ▷ See Sec. 4.2.1
16:     **for** $s = 1 \dots S_{\text{PPO}}$ **do**
            ▷ *See Sec. 4.2 for details.*
17:         Compute PPO unlearning objective $\mathcal{J}(\theta)$ on $B_f$.
18:         Compute distillation regularization $\mathcal{L}_{\text{dis}}(\theta)$ on $B_r$.
19:         Update $\theta \leftarrow \theta + \eta_\theta \nabla_\theta (\mathcal{J}(\theta) - \lambda_{\text{dis}} \mathcal{L}_{\text{dis}}(\theta))$.
20:     **end for**
21: **end for**
22: **return** $\pi$

---

tive leads to less over-forgetting (e.g., generation of gibberish text) than the GA objective. As with the prediction loss-based objectives, we cannot directly show that the de-attribution objective implies perfect unlearning. Thus, in App. C, we show that lower attribution scores correlate with more unlearning.

**Practical consideration of attribution functions.** Ideally, the formulation Eq. (2) relies on an ideal attribution function $\mathtt{DA}^*$. In practice, however, due to computation constraints/limitations on existing data attribution methods, $\mathtt{DA}$ might not be perfect. Moreover, since the core motivation for LLM unlearning is to avoid the prohibitive cost of fully retraining the LLM, computational efficiency is a key constraint for attribution functions. Specifically, the computation cost of $\mathtt{DA}$ should be minimal given that it is invoked iteratively during the unlearning process. While

there have been several categories of existing attribution functions, including leave-one-out (LOO), Shapley value, datamodeling, influence function, and text watermarking, the high computational costs make many of them impractical in the LLM unlearning setting. Specifically, LOO requires retraining, which defeats the purpose of unlearning; Shapley value and influence function approaches incur high computational costs (Hammoudeh & Lowd, 2024), rendering them impractical for LLM unlearning. While *text watermarking* methods usually offer reliable and efficient verification, it tediously requires all training data to be watermarked in advance, which might not be practical in certain conditions. We provide a detailed discussion on existing attribution functions in App. A.

Consequently, we propose an efficient approximation of the attribution function: After training and before unlearning, we train a **lightweight classifier** $\mathtt{DA}_c$ to predict the owner whose dataset most likely influences the given generation $y_d$, as shown in Algo. 1 Phase 1. We then derive the attribution score directly from the prediction probability $\mathtt{DA}_c(i, y_d) = p(i \mid y_d)$. Although this classifier may not be a perfect approximation of an ideal attribution function, it offers the advantage of efficiency, as training $\mathtt{DA}_c$ is efficient and offline before the unlearning process. Additionally, $\mathtt{DA}_c$ remains frozen during unlearning and can be discarded after the completion of unlearning.

**Implementation of $\mathtt{DA}_c$.** We implement the attribution classifier $\mathtt{DA}_c$ by attaching a classification head to a lightweight LLM with Low-Rank Adaptation (LoRA). Specifically, we freeze the original LLM parameters and update only the LoRA adapters and classification head on $\mathcal{D}_{\mathcal{N}}$, where samples $y_d^*$ from each subset $\mathcal{D}_i$ are assigned the label $i$. This design efficiently leverages the pre-trained LLM's ability to capture the semantic information for each subset $\mathcal{D}_i$. Additionally, we have also explored using a sentence transformer to perform classification directly on sentence embeddings. In the rest of this paper, we focus on this efficient LLM classifier-based technique and compare the implementation of $\mathtt{DareU}$ using other attribution functions in Sec. 5.3.

*Remark.* Our implementation above is efficient and sufficient to demonstrate the effectiveness of $\mathtt{DareU}$ in Sec. 5.2. If there is a better attribution function, it can be adopted as well and will likely improve the effectiveness of $\mathtt{DareU}$.

### 4.2. Formulating Unlearning with PPO

Since our data attribution-based formulation naturally implies an explicit reward model, we employ reward-based RL algorithms to effectively align the LLM-generated re-

sponses and minimize the attributability to the forget set. Specifically, we adopt PPO, a standard reward-based RL method often used in LLM post-training, to update the LLM using advantages estimated from attribution scores and ensure training stability by preventing large updates. We empirically demonstrate that by controlling the updates via clipping and regularization, our $\mathtt{DareU}$ using PPO with attribution rewards achieves effective unlearning by balancing between forget quality and model utility. We provide the background of PPO in App. D. The PPO algorithm design for effective LLM unlearning is detailed below with the pseudocode provided in Algo. 1.

Consider a text sample $d$. When the unlearned LLM at the start of a data epoch $\pi_{\mathrm{old}}$ (see Algo. 1) is queried with $q_d$, it generates a response $y_d = (y_{d[t]})_{t=0}^{T-1}$ with $T$ tokens. Let $y_{d[:t]}$ denote the first $t$ tokens in $y_d$ and let the context $c_{t,d}$ denote the concatenation of $q_d$ and $y_{d[:t]}$. At each token position $t$, we define the advantage $\hat{A}_{t,d}$ and reward signal $\hat{R}_{t,d}$ associated with $y_d$ as:

$$\hat{A}_{t,d} = \hat{R}_{t,d} - V(c_{t,d}), \qquad \hat{R}_{t,d} = \sum_{j=0}^{T-t-1} \gamma^j r_{t+j,d},$$

where $r_{t,d}$ is the token-level reward (to be defined in Sec. 4.2.1), $\gamma \in (0,1)$ is the discount factor, and $V$ is a learned value function that estimates the expected discounted reward from context $c_{t,d}$ under the policy $\pi$. In PPO, we consider the next token (action) probability given the current context from the current LLM $\pi$ and compare it with the LLM at the start of the data epoch $\pi_{\mathrm{old}}$. At token position $t$, the policy ratio associated with $y_d$ is defined as $s_{t,d}(\theta) := \frac{\pi(y_{d[t]}|c_{t,d})}{\pi_{\mathrm{old}}(y_{d[t]}|c_{t,d})}$ where $\pi(y_{d[t]} \mid c_{t,d})$ denotes the probability of $\pi$ generating token $y_{d[t]}$ given context $c_{t,d}$. The PPO unlearning objective $\mathcal{J}(\theta)$ is to maximize the expected advantage over the forget set $\mathcal{D}_{\mathcal{F}}$. For each sample $d \sim \mathcal{D}_{\mathcal{F}}$, the expectation is taken over each token position $t$ in the LLM-generated response $y_d$:

$$\mathbb{E}_{d \sim \mathcal{D}_{\mathcal{F}}} \mathbb{E}_t \Big[ \min \big( s_{t,d}(\theta) \hat{A}_{t,d},$$
$$\mathrm{clip}(s_{t,d}(\theta), 1 - \varepsilon, 1 + \varepsilon) \hat{A}_{t,d} \big) \Big],$$

where $\varepsilon > 0$ is the clipping parameter to prevent large policy/LLM updates.

Note that in our LLM unlearning setting, only forget set advantages contribute to the PPO unlearning objective $\mathcal{J}(\theta)$. To consider the retain set $\mathcal{D}_{\mathcal{R}}$ in our objective and preserve model utility, we introduce a distillation regularization term inspired by policy distillation (Rusu et al., 2015). Specifically, we align the current LLM with the reference LLM (i.e., the original LLM) $\pi_{\mathcal{N}}$ over the retain set $\mathcal{D}_{\mathcal{R}}$ by considering the KL divergence of the LLMs' probabilities at each token position $t$ in each generated response:

$$\mathcal{L}_{\mathrm{dis}} := \mathbb{E}_{d \sim \mathcal{D}_{\mathcal{R}}} \mathbb{E}_t \Big[ \mathrm{KL}\big( \pi_{\mathcal{N}}(y_{d[t]} \mid c_{t,d}) \parallel \pi(y_{d[t]} \mid c_{t,d}) \big) \Big].$$

---

[2]In our implementation, we loop through $\mathcal{D}_{\mathcal{N}}$ once (we use data epoch = 1). With more data epochs (in App. F.3), $\pi_{\mathrm{old}}$ is updated after each data epoch.

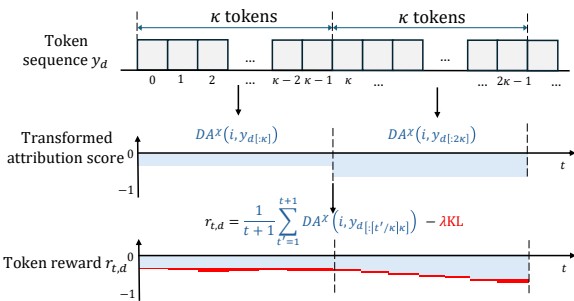

*Figure 3.* Visualization of how the token reward $r_{t,d}$'s are computed for various token position $t$. The attribution function DA is only invoked and transformed every $\kappa$ tokens. The reward $r_{t,d}$ is the sum of the cumulative mean of the transformed attribution scores and the negated KL divergence between the token distribution of the unlearned and previous epoch LLM.

Finally, we formulate the overall training objective by incorporating the distillation regularization term:

$$\max_{\theta} \; \mathcal{J}(\theta) \; - \; \lambda_{\text{dis}} \, \mathcal{L}_{\text{dis}}(\theta),$$

where $\lambda_{\text{dis}} \geq 0$ is a hyperparameter controlling the strength of the distillation regularization. The first term $\mathcal{J}(\theta)$ drives the unlearning process by de-attributing to $\mathcal{D}_{\mathcal{F}}$, and the second term $\mathcal{L}_{\text{dis}}(\theta)$ preserves the original behavior of $\pi_{\mathcal{N}}$ on $\mathcal{D}_{\mathcal{R}}$. Therefore, our DareU ensures a balance between forget quality and model utility. In practice, DareU alternates between these forget updates and retain updates, similar to that in SCRUB (Kurmanji et al., 2023).

### 4.2.1. Practical Computation of Reward Signals

How do we set $r_{t,d}$? Suppose that $d$ belongs to data owner $i$. $r_{t,d}$ should depend on the attribution score $\text{DA}(i, y_{d[:t]})$ of the LLM-generated response up to the $t$ token. However, as the attribution function may be as expensive as querying an LLM, it is computationally expensive to invoke it for every token in $d$ and for every $d \in \mathcal{D}_{\mathcal{F}}$. We propose to reduce the computation cost as shown in Fig. 3 and described as follows. For every $d$, the attribution function is only invoked every $\kappa$ tokens, i.e., using the prefixes $(y_{d[:k\kappa]})_{k=1}^{\lceil T/\kappa \rceil}$ to obtain the attribution score, e.g, $\text{DA}(i, y_{d[:k\kappa]})$. The attribution score is then broadcast to all these $\kappa$ tokens. Subsequently, to reduce the high variance across every $\kappa$ tokens and to propagate the early reward signals forward along the sequence, we compute a prefix-averaged base reward by taking the cumulative mean over token positions:

$$r_{t,d} := \frac{1}{t+1} \sum_{t'=1}^{t+1} \text{DA}^{\chi}(i, y_{d[:\lceil t'/\kappa \rceil \kappa]})$$
$$- \lambda \text{KL}\big(\pi(y_{d[t]} \mid c_{t,d}) \,\|\, \pi_{\text{old}}(y_{d[t]} \mid c_{t,d})\big),$$

where the attribution scores are transformed through a log-based function $\chi$ to improve stability/performance. To prevent the unlearning process from drastically changing the

LLM and degrading general model utility, we incorporate a token-level KL divergence regularization where $\lambda$ controls the strength of the regularization. We empirically validate the effectiveness of our design by comparing it with the sequence-level and per-token rewards in App. F.1.

**Transformed attribution score.** To improve the stability of the reward, inspired by Schulman et al. (2017), we apply a log-based transformation $\chi$ : $b = \text{clip}(\ln(1 - \text{clip}(b, \varepsilon, \, 1-\varepsilon))/C_{\text{seg}}, -1, 0)$ to the attribution scores $\text{DA}(i, y_{d[:t]})$, i.e., $\text{DA}^{\chi}(i, y_{d[:t]}) = \chi(\text{DA}^{\chi}(i, y_{d[:t]}))$ to map the probability within $[-1, 0]$. Details on the transformation are explained in App. D. A lower attribution score corresponds to a higher reward, thus the objective is to maximize the expected advantage over $\mathcal{D}_{\mathcal{F}}$.

## 5. Experiments

### 5.1. Experimental Setup

**Datasets and Models.** We consider two unlearning benchmark datasets with different tasks: (1) **TOFU** dataset (Maini et al., 2024) for the question-answering task contains 4000 question-answer pairs generated by GPT-4 with an average length of 60 tokens. We unlearn 10% of the dataset (400 samples). (2) **ArXiv** dataset (Lu et al., 2026) for text completion task comprises 8000 arXiv paper abstracts from 20 academic paper categories with an average length of 223 tokens. We unlearn the last category in the dataset (400 samples). We fine-tune Llama2-7B-chat (Touvron et al., 2023) and Qwen3-8B (Yang et al., 2025) with LoRA adapters (Hu et al., 2022) as the base LLMs for unlearning. We then fine-tune TinyLlama (Zhang et al., 2024a) and Qwen3-0.6B with LoRA and a classification head as the data attribution classifier. Training hyperparameters are detailed in App. E.1.

**Unlearning Baselines.** We compare our DareU with baseline unlearning approaches including: **Retraining** as the gold standard; prediction loss-based approaches such as **Fine-tuning**, **Gradient Ascent** (GA), **GDiff** (Maini et al., 2024), **SCRUB** (Kurmanji et al., 2023), and a second-order approach **SCuReNewton** (SCRN) (Bui et al., 2026); as well as preference optimization baselines such as Direct Preference Optimization that learns "**I don't know**" as the forget set response (IDK) (Maini et al., 2024) and **Negative Preference Optimization** (NPO) (Zhang et al., 2024b) that discourages the LLM from generating the forget set's target responses. Detailed descriptions of the baselines and hyperparameter tuning are in App. E.2.

**Evaluation Metrics.** To assess the effectiveness of unlearning, we follow previous works (Maini et al., 2024) and compute the **ROUGE-L Recall** scores on $\mathcal{D}_{\mathcal{F}}$, $\mathcal{D}_{\mathcal{R}}$, and $\mathcal{D}_{test}$[3], where $\mathcal{D}_{\mathcal{F}}$ ROUGE measures forget quality and $\mathcal{D}_{\mathcal{R}}$

---

[3]As the ArXiv dataset has no test set, we omit its $\mathcal{D}_{test}$ ROUGE

*Table 2.* Unlearning performance on Llama2-7B on TOFU and Arxiv datasets (over 3 random runs). We use "→" to denote closer to Retraining is better and "↑" to denote higher is better. The best results are **bold** and the second-best are underlined.

| Approach | TOFU | | | | | | ArXiv | | | | | |
|---|---|---|---|---|---|---|---|---|---|---|---|---|
| | $\mathcal{D}_\mathcal{F}$ ROUGE (→) | $\mathcal{D}_\mathcal{R}$ ROUGE (→) | $\mathcal{D}_{test}$ ROUGE (→) | ToW (↑) | Truth Ratio (↑) | MIA (→) | $\mathcal{D}_\mathcal{F}$ ROUGE (→) | $\mathcal{D}_\mathcal{R}$ ROUGE (→) | ToW* (↑) | $\mathcal{D}_\mathcal{F}$ ES (↓) | $\mathcal{D}_\mathcal{R}$ ES (↑) | MIA (→) |
| Original | 0.908 ± 0.007 | 0.901 ± 0.003 | 0.812 ± 0.014 | 0.455 ± 0.010 | 0.508 ± 0.001 | 1.0000 ± 0.0000 | 0.825 ± 0.000 | 0.895 ± 0.000 | 0.342 ± 0.000 | 0.101 ± 0.023 | 0.205 ± 0.002 | 0.9106 ± 0.0000 |
| Retraining | 0.381 ± 0.004 | 0.917 ± 0.009 | 0.818 ± 0.030 | 1.000 ± 0.000 | 0.669 ± 0.002 | 0.9971 ± 0.0025 | 0.172 ± 0.000 | 0.908 ± 0.000 | 1.000 ± 0.000 | 0.000 ± 0.000 | 0.198 ± 0.017 | 0.5796 ± 0.0000 |
| Fine-tune | 0.772 ± 0.005 | **0.887 ± 0.004** | 0.775 ± 0.013 | 0.563 ± 0.017 | 0.510 ± 0.004 | 0.9998 ± 0.0002 | 0.646 ± 0.081 | **0.763 ± 0.055** | 0.482 ± 0.052 | 0.062 ± 0.011 | **0.170 ± 0.028** | 0.8983 ± 0.0148 |
| GA | 0.000 ± 0.000 | 0.001 ± 0.001 | 0.000 ± 0.000 | 0.009 ± 0.001 | 0.739 ± 0.029 | 0.9990 ± 0.0013 | 0.197 ± 0.018 | 0.283 ± 0.026 | 0.439 ± 0.015 | 0.006 ± 0.008 | 0.030 ± 0.019 | 0.6515 ± 0.0410 |
| GDiff | 0.419 ± 0.045 | 0.601 ± 0.074 | 0.760 ± 0.026 | 0.614 ± 0.057 | 0.541 ± 0.010 | 0.9998 ± 0.0003 | 0.243 ± 0.108 | 0.442 ± 0.083 | 0.532 ± 0.055 | 0.001 ± 0.001 | 0.081 ± 0.004 | 0.8183 ± 0.1621 |
| SCRUB | 0.268 ± 0.226 | 0.302 ± 0.266 | 0.283 ± 0.364 | 0.219 ± 0.265 | 0.738 ± 0.149 | 0.6640 ± 0.4695 | 0.024 ± 0.028 | 0.432 ± 0.077 | 0.511 ± 0.052 | **0.000 ± 0.000** | 0.067 ± 0.011 | 0.0257 ± 0.0015 |
| SCRN | 0.292 ± 0.135 | 0.270 ± 0.115 | 0.657 ± 0.461 | 0.251 ± 0.161 | **0.774 ± 0.028** | **0.9967 ± 0.0032** | **0.196 ± 0.141** | 0.191 ± 0.136 | 0.312 ± 0.122 | **0.000 ± 0.000** | 0.009 ± 0.013 | **0.5797 ± 0.0688** |
| IDK | 0.028 ± 0.003 | 0.804 ± 0.012 | 0.795 ± 0.026 | 0.548 ± 0.007 | 0.530 ± 0.004 | 1.0000 ± 0.0001 | - | - | - | - | - | - |
| NPO | 0.286 ± 0.018 | 0.318 ± 0.011 | 0.473 ± 0.102 | 0.239 ± 0.047 | 0.596 ± 0.011 | 0.9903 ± 0.0134 | 0.522 ± 0.021 | 0.642 ± 0.039 | 0.528 ± 0.047 | 0.039 ± 0.016 | 0.154 ± 0.022 | 0.7774 ± 0.0164 |
| DareU | **0.407 ± 0.015** | 0.797 ± 0.006 | **0.834 ± 0.030** | **0.816 ± 0.011** | 0.564 ± 0.005 | 0.9962 ± 0.0042 | 0.456 ± 0.004 | 0.704 ± 0.012 | **0.622 ± 0.005** | 0.013 ± 0.007 | 0.127 ± 0.016 | 0.8801 ± 0.0072 |

*Table 3.* Unlearning performance on Qwen3-8B on TOFU and Arxiv datasets (over 3 random runs). We use "→" to denote closer to Retraining is better and "↑" to denote higher is better. The best results are **bold** and the second-best are underlined.

| Approach | TOFU | | | | | | ArXiv | | | | | |
|---|---|---|---|---|---|---|---|---|---|---|---|---|
| | $\mathcal{D}_\mathcal{F}$ ROUGE (→) | $\mathcal{D}_\mathcal{R}$ ROUGE (→) | $\mathcal{D}_{test}$ ROUGE (→) | ToW (↑) | Truth Ratio (↑) | MIA (→) | $\mathcal{D}_\mathcal{F}$ ROUGE (→) | $\mathcal{D}_\mathcal{R}$ ROUGE (→) | ToW* (↑) | $\mathcal{D}_\mathcal{F}$ ES (↓) | $\mathcal{D}_\mathcal{R}$ ES (↑) | MIA (→) |
| Original | 0.848 ± 0.008 | 0.808 ± 0.008 | 0.664 ± 0.039 | 0.501 ± 0.017 | 0.378 ± 0.000 | 1.0000 ± 0.0000 | 0.779 ± 0.000 | 0.835 ± 0.000 | 0.392 ± 0.006 | 0.071 ± 0.008 | 0.126 ± 0.028 | 1.0000 ± 0.0000 |
| Retraining | 0.375 ± 0.002 | 0.827 ± 0.014 | 0.687 ± 0.040 | 1.000 ± 0.000 | 0.537 ± 0.001 | 0.9967 ± 0.0009 | 0.170 ± 0.000 | 0.846 ± 0.000 | 1.000 ± 0.000 | 0.000 ± 0.000 | 0.145 ± 0.033 | 0.6814 ± 0.0000 |
| Fine-tune | 0.727 ± 0.015 | **0.827 ± 0.016** | 0.710 ± 0.024 | 0.702 ± 0.034 | 0.390 ± 0.006 | 1.0000 ± 0.0000 | 0.494 ± 0.010 | 0.582 ± 0.016 | 0.530 ± 0.008 | 0.093 ± 0.020 | **0.231 ± 0.043** | 0.8636 ± 0.1900 |
| GA | 0.003 ± 0.002 | 0.003 ± 0.001 | 0.000 ± 0.000 | 0.028 ± 0.003 | **0.512 ± 0.035** | 0.3829 ± 0.1807 | 0.404 ± 0.291 | 0.458 ± 0.327 | 0.385 ± 0.159 | 0.006 ± 0.008 | 0.030 ± 0.019 | 0.7824 ± 0.3077 |
| GDiff | 0.281 ± 0.190 | 0.347 ± 0.203 | 0.527 ± 0.298 | 0.406 ± 0.253 | 0.414 ± 0.076 | 0.7029 ± 0.4039 | **0.207 ± 0.116** | 0.430 ± 0.111 | 0.558 ± 0.120 | 0.001 ± 0.001 | 0.081 ± 0.004 | 0.8636 ± 0.1900 |
| SCRUB | 0.498 ± 0.063 | 0.644 ± 0.026 | **0.689 ± 0.055** | 0.731 ± 0.015 | 0.425 ± 0.011 | 0.9942 ± 0.0039 | 0.023 ± 0.028 | 0.346 ± 0.021 | 0.465 ± 0.028 | **0.000 ± 0.000** | 0.067 ± 0.011 | 0.0254 ± 0.0043 |
| SCRN | 0.573 ± 0.383 | 0.554 ± 0.369 | 0.457 ± 0.323 | 0.409 ± 0.261 | 0.262 ± 0.159 | 0.9453 ± 0.0773 | 0.337 ± 0.049 | 0.380 ± 0.094 | 0.481 ± 0.049 | 0.067 ± 0.054 | 0.132 ± 0.095 | 0.9972 ± 0.0030 |
| IDK | 0.191 ± 0.013 | 0.754 ± 0.013 | 0.705 ± 0.031 | 0.668 ± 0.019 | 0.406 ± 0.004 | 1.0000 ± 0.0000 | - | - | - | - | - | - |
| NPO | 0.640 ± 0.023 | 0.637 ± 0.025 | 0.704 ± 0.035 | 0.638 ± 0.021 | 0.472 ± 0.020 | 0.9892 ± 0.0147 | 0.309 ± 0.033 | 0.387 ± 0.041 | 0.506 ± 0.031 | 0.039 ± 0.016 | 0.154 ± 0.022 | 0.9965 ± 0.0018 |
| DareU | **0.465 ± 0.010** | 0.718 ± 0.004 | 0.670 ± 0.049 | **0.797 ± 0.009** | 0.406 ± 0.007 | 0.9999 ± 0.0001 | 0.405 ± 0.020 | **0.659 ± 0.005** | **0.660 ± 0.018** | 0.025 ± 0.015 | 0.090 ± 0.028 | 0.9998 ± 0.0002 |

and $\mathcal{D}_{test}$ ROUGE measure model utility. Since the goal of LLM unlearning is to approximate the retrained LLM (Sec. 3), we compare ROUGE between the unlearned LLM and the retrained LLM (closer to Retraining is better). We also report the **Tug-of-War** (ToW) score defined in (Zhao et al., 2024) to aggregate the ROUGE gaps across $\mathcal{D}_\mathcal{F}$, $\mathcal{D}_\mathcal{R}$, and $\mathcal{D}_{test}$ to demonstrate the balance between forget quality and model utility (higher is better). To evaluate unlearning from other perspectives, for TOFU, we compute the **Truth Ratio** (Maini et al., 2024) which measures the probability of the unlearned LLM generating correct versus incorrect answers (higher is better). For ArXiv, we compute the **Extraction Strength** (ES) implemented in (Wang et al., 2025b), which evaluates memorization by measuring the minimal prefix length to reproduce the suffix (lower is better). We additionally compute the AUC of the **Membership Inference Attack** (MIA) using Min-40% attack (Shi et al., 2024a) to evaluate unlearning from the privacy perspective.

## 5.2. Main Experiment Results

**Unlearning Effectiveness.** We first compare the unlearning performance of DareU and the other baseline approaches. As shown in Table 2 and Table 3, DareU achieves effective unlearning especially on TOFU, where it achieves the best forget quality by reaching the closest $\mathcal{D}_\mathcal{F}$ ROUGE to Retraining, and preserves the model utility by maintaining high ROUGE scores on $\mathcal{D}_\mathcal{R}$ and $\mathcal{D}_{test}$. More importantly, DareU achieves **the highest ToW score** across different datasets and LLMs, outperforming all baseline approaches, which indicates that DareU achieves the best balance be-

and compute a modified ToW* without $\mathcal{D}_{test}$.

tween forget quality and model utility. While the performance inevitably drops on ArXiv due to longer sequence lengths and larger dataset size, which presents a more challenging unlearning task, DareU maintains comparable forget quality to strong baselines while achieving the best balance between forget quality and model utility. In contrast, many prediction loss-based optimization approaches either suffer from over-forgetting with degraded performance on $\mathcal{D}_\mathcal{R}$ (e.g., GA, GDiff, SCRUB) or fail to effectively unlearn $\mathcal{D}_\mathcal{F}$ (e.g., Fine-tune). Notably, IDK produces a $\mathcal{D}_\mathcal{F}$ ROUGE score much lower than that of Retraining because it forces the LLM to respond with refusals like "I don't know" on $\mathcal{D}_\mathcal{F}$ rather than approximating an LLM that was never trained on $\mathcal{D}_\mathcal{F}$. We empirically observe that MIA is less informative on TOFU where even Retraining achieves high MIA, likely due to the distributional difference between forget and hold-out sets (Bui et al., 2026). Overall, our DareU consistently achieves state-of-the-art or comparable results across different evaluation metrics, demonstrating the effectiveness of our DareU. In App. F.6, we also demonstrate that DareU leads to a lower percentage of incoherent responses, thus addressing (**L1**) in Sec. 3.

**Unlearning Efficiency.** One potential drawback of adopting PPO for LLM unlearning is that it may incur higher computational costs than prediction loss-based approaches due to its token-by-token autoregressive generations. We compare the total unlearning time and peak memory usage of DareU against baseline unlearning approaches in Table 4 supported by a formal time complexity analysis in App. G. Empirically, DareU incurs a longer unlearning time than many baseline approaches, but the time is comparable to second-order approaches such as SCRN. Although DareU requires more

*Table 4.* Unlearning efficiency across different datasets and models (over 3 random runs) measured by average unlearning time and peak memory usage (lower is better).

| Approach | Llama2 × TOFU | | Llama2 × ArXiv | | Qwen3 × TOFU | | Qwen3 × ArXiv | |
|---|---|---|---|---|---|---|---|---|
| | Time/s | Mem./MB | Time/s | Mem./MB | Time/s | Mem./MB | Time/s | Mem./MB |
| Retraining | 1223.5 | 55331 | 7852.5 | 55331 | 1499.6 | 75148 | 9656.7 | 75148 |
| Fine-tune | 254.2 | 55331 | 524.4 | 55331 | 311.7 | 75148 | 652.2 | 75148 |
| GA | **30.8** | 55331 | 29.83 | 55331 | 37.3 | 75148 | 37.9 | 75148 |
| GDiff | 55.4 | 96267 | 108.8 | 96267 | 132.0 | 128422 | 69.2 | 128422 |
| SCRUB | 257.4 | 70521 | 258.1 | 70521 | 517.4 | 101542 | 308.0 | 101542 |
| SCRN | 539.7 | 112625 | 457.9 | 112625 | 1550.5 | 132736.7 | 1598.8 | 132736.7 |
| IDK | 254.2 | 96970 | - | - | 339.7 | 75148 | - | - |
| NPO | 136.6 | 70042 | 48.8 | 70042 | 73.6 | 99195 | 74.1 | 99195 |
| DareU | 471.3 | 69239.9 | 1595.3 | 122859.0 | 521.0 | 97676.0 | 1850.7 | 133286.6 |

*Table 5.* Data attribution accuracy of different methods (higher is better) measured on the TOFU dataset.

| Method | WASA | ST Classifier | LM Classifier (ours) |
|---|---|---|---|
| Accuracy | 0.794 ± 0.021 | 0.878 ± 0.044 | 0.877 ± 0.008 |

computational cost on the ArXiv dataset, where significantly longer sequence lengths result in longer autoregressive generation time, `DareU` remains significantly more efficient than Retraining. We further note that while some baseline approaches like GA offer advantages in efficiency, they often fail to achieve satisfactory unlearning performance. Overall, we argue that our `DareU` offers a justified trade-off that users can choose: Table 2 and Table 3 demonstrate that the higher computational cost leads to better unlearning performance, with balanced forget quality and model utility, than other faster but less reliable baseline unlearning approaches. Moreover, `DareU` computational costs can be further reduced by integrating high-throughput inference engines (e.g., vLLM) to accelerate LLM generations.

### 5.3. Choice of Attribution Functions

In this section, we present a comprehensive analysis of the choice of attribution functions $DA_c$ by comparing our proposed lightweight LLM-based classifier (LM classifier) against other efficient data attribution methods, including using the sentence transformer-based classifier (ST classifier) and text watermarking, as discussed in Sec. 4.1. For the sentence transformer-based classifier, we adopt DeBER-TaV3 (He et al., 2023) and train it to perform classification in a similar manner to that described in Sec. 4.1. For text watermarking, we adopt a recent approach designed for data attribution: WASA (Lu et al., 2025) proposes to embed Unicode characters into the training texts as watermarks such that during verification, the presence of the watermark attributes the generated response to the corresponding data owner. We obtain the attribution function by computing the prediction probability for the corresponding watermark.

**Data Attribution Accuracy.** We first evaluate the data attribution accuracy of different methods by applying them to responses $y_d$ generated by $\pi_\mathcal{N}$ given queries $q_d$. While it is hard to attribute the actual ground truth source $i$ for

*Table 6.* Unlearning performance of `DareU` when implemented with different attribution functions, measured on Llama2 × TOFU.

| Approach | TOFU | | | | | |
|---|---|---|---|---|---|---|
| | $\mathcal{D}_\mathcal{F}$ ROUGE | $\mathcal{D}_\mathcal{R}$ ROUGE | $\mathcal{D}_{test}$ ROUGE | ToW | Truth Ratio | MIA |
| Original | 0.908 ± 0.007 | 0.901 ± 0.003 | 0.812 ± 0.014 | 0.455 ± 0.010 | 0.508 ± 0.001 | 1.0000 ± 0.0000 |
| Retraining | 0.381 ± 0.004 | 0.917 ± 0.009 | 0.818 ± 0.030 | 1.000 ± 0.000 | 0.669 ± 0.002 | 0.9971 ± 0.0025 |
| WASA | 0.427 ± 0.009 | 0.792 ± 0.002 | 0.798 ± 0.011 | 0.806 ± 0.010 | 0.558 ± 0.008 | 0.9984 ± 0.0019 |
| ST Classifier | 0.426 ± 0.009 | **0.798 ± 0.008** | 0.780 ± 0.024 | 0.800 ± 0.037 | **0.564 ± 0.003** | 0.9994 ± 0.0003 |
| LM Classifier | **0.407 ± 0.015** | 0.797 ± 0.006 | **0.834 ± 0.030** | 0.816 ± 0.011 | **0.564 ± 0.005** | 0.9962 ± 0.0042 |

*Table 7.* Cross-validation of `DareU` implemented with different attribution functions using multiple attribution metrics, measured on Llama2 × TOFU.

| Method | WASA metric | ST Classifier metric | LM Classifier metric |
|---|---|---|---|
| Original | 0.854 ± 0.069 | 0.820 ± 0.074 | 0.738 ± 0.007 |
| WASA | 0.772 ± 0.045 | 0.481 ± 0.091 | 0.376 ± 0.015 |
| ST Classifier | 0.702 ± 0.056 | 0.450 ± 0.063 | 0.396 ± 0.045 |
| LM Classifier | 0.703 ± 0.037 | 0.423 ± 0.065 | 0.324 ± 0.009 |

an LLM-generated response $y_d$ (e.g., it may require human judgment to identify the true source), we assume that the source of $y_d$ corresponds to the source of $q_d$ following (Lu et al., 2025) given the high response accuracy of $\pi_\mathcal{N}$ (suggested by high ROUGE scores). As shown in Table 5, while our LM classifier and other attribution approaches are efficient approximations of an ideal attribution function, they still achieve high attribution accuracy and are sufficient to demonstrate the effectiveness of `DareU`. In App. F.5.1, we further show that `DareU` is still effective and robust to inaccuracies in or overfitting to the classifier by considering adding noise to the classifier or using another held-out classifier.

**Unlearning Effectiveness.** Table 6 compares the unlearning performance of `DareU` implemented with different attribution functions, which shows that they all yield comparably effective unlearning results. This validates the effectiveness of adopting data de-attribution as an unlearning objective.

**Cross Validation.** With these attribution functions as metrics, we further cross-validate the effectiveness of `DareU` to demonstrate that `DareU` achieves genuine unlearning rather than learning to exploit the specific attribution function. Specifically, in Table 7, we evaluate `DareU` implemented with different attribution functions using hold-out attribution metrics, e.g., using the ST classifier or WASA to assess `DareU` implemented with the LM classifier. The results illustrate that `DareU` reduces attribution scores across hold-out metrics, thus validating that `DareU` genuinely de-attributes the forget set.

### 5.4. Ablation Studies

To further demonstrate the effectiveness and robustness of `DareU` under different unlearning scenarios, we conduct the following ablation studies:

**Unlearning Scalability.** We evaluate the scalability of `DareU` by assessing its performance for both smaller and

larger $\mathcal{D}_{\mathcal{F}}$ using TOFU. Results in Table 11 in App. F.2 demonstrate that `DareU` achieves effective unlearning performance and balances well between forget quality and model utility across different sizes of $\mathcal{D}_{\mathcal{F}}$.

**Sensitivity to Hyperparameters.** We investigate the sensitivity of `DareU` with respect to hyperparameter selection through extensive ablation studies presented in App. F.3. Experimental results in Tables 12-16 that vary hyperparameters such as learning rate, $\kappa$, $\lambda_{\mathrm{dis}}$, etc., demonstrate that `DareU` consistently achieves effective unlearning performance. This validates that `DareU` is robust to hyperparameter variations within reasonable ranges and requires minimal hyperparameter tuning in practice.

**Sequential Unlearning.** We further consider sequential unlearning in App. F.4 as a more challenging setting for LLM unlearning, where the full $\mathcal{D}_{\mathcal{F}}$ is unlearned sequentially over multiple unlearning rounds. As visualized in Fig. 6, our `DareU` balances forget quality and model utility well and achieves performance close to Retraining over 5 unlearning rounds during sequential unlearning.

**Unlearning Robustness.** To verify that `DareU` achieves genuine unlearning rather than simply masking information through overfitting to specific forget queries or reward hacking without genuinely removing the content from $\mathcal{D}_{\mathcal{F}}$, we conduct the following robustness evaluations in App. F.5 and stress-test the ability of `DareU` to: **(a)** perform effective unlearning even with inaccurate attribution functions (App. F.5.1); **(b)** perform effective unlearning in the presence of **semantic overlaps** in $\mathcal{D}_{\mathcal{F}}$ and $\mathcal{D}_{\mathcal{R}}$ that decreases the classifier accuracy (App. F.5.2); **(c)** resist **adversarial probing** via paraphrased forget queries (App. F.5.3); and **(d)** prevent rapidly recovering the unlearned knowledge through **relearning** $\mathcal{D}_{\mathcal{F}}$ (App. F.5.4). Empirical results show that `DareU` achieves consistent performance even under these tests, which validates its genuine unlearning ability.

## 6. Conclusion

In this work, we introduce a novel perspective and optimization objective for LLM unlearning. De-attributing the LLM-generated responses to the forget set is a more precise objective and offers consistent target values. We propose `DareU`, which achieves this objective effectively and stably by using PPO. Experiment results show that our proposed `DareU` outperforms existing baselines and balances between forget quality and model utility. While we adopted a simple classifier as an efficient approximation of an ideal attribution function to demonstrate the effectiveness of `DareU`, future advances in data attribution or reinforcement learning may improve its efficiency and unlearning performance.

## Limitations

While this work introduces data de-attribution as a novel objective for LLM unlearning, we acknowledge the following limitations:

- As shown in Table 4, `DareU` may incur higher computational costs than simpler prediction loss-based unlearning approaches. Despite this, `DareU` demonstrates better unlearning effectiveness in Sec. 5.2 and will be preferred over prediction loss-based approaches when (i) better unlearning performance that balances forget quality and model utility is desired, and (ii) the users have a higher computational budget and optionally the ability to exploit high-throughput inference engines to accelerate LLM generations. The user is free to choose from the trade-off based on their computation budget.
- Future work should develop better and more efficient attribution functions. We will include more failure modes of `DareU`, such as inaccurate classifiers.
- The practitioner must have access to the pretraining or finetuning dataset in order to obtain a data attribution function or classifier. While this might exclude the application of `DareU` for ensuring successful pre-training data removal, `DareU` is still useful for many other real-world applications, such as when healthcare firms have to unlearn private health data that they finetune their LLM on.

## Impact Statement

This paper introduces a novel LLM unlearning approach to effectively remove the influence of some unwanted data from a trained LLM. This hence suggests potential social impacts on data privacy/copyright protection by removing a data owner's data and its influence from a trained LLM upon request, adhering to regulations such as "right to be forgotten" in the era of LLMs. Meanwhile, users of DareU should be cautious that data de-attribution may not certify unlearning or data removal compliance and should instead verify unlearning through other methods. The attribution classifier should be protected to prevent adversarial uses such as membership inference.

## Acknowledgments

This research is supported by the National Research Foundation Singapore and the Singapore Ministry of Digital Development and Innovation, National AI Group under the AI Visiting Professorship Programme (award number AIVP-2024-001). This research is also supported by the National Research Foundation, Singapore under its AI Singapore Programme (AISG Award No: AISG3-RP-2022-029).

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

*Table 8.* Full-parameter unlearning performance of `DareU` on full-parameter Llama2 $\times$ TOFU without LoRA. We only report the results on seed 42 due to high computational costs.

| Approach | $\mathcal{D}_{\mathcal{F}}$ ROUGE ($\rightarrow$) | $\mathcal{D}_{\mathcal{R}}$ ROUGE ($\rightarrow$) | $\mathcal{D}_{test}$ ROUGE ($\rightarrow$) | ToW ($\uparrow$) | Truth Ratio ($\uparrow$) | MIA ($\rightarrow$) |
|---|---|---|---|---|---|---|
| Original | 0.932 | 0.926 | 0.929 | 0.379 | 0.401 | 0.996 |
| Retraining | 0.341 | 0.994 | 0.972 | 1.000 | 0.995 | 0.962 |
| `DareU` | 0.288 | 0.793 | 0.778 | 0.752 | 0.768 | 0.978 |

## A. LLM Data Attribution

Data attribution for LLM-generated responses aims to identify whether a specific training subset (data owner) influences those generations. Conventional data attribution methods can be categorized into **retraining-based** and **gradient-based** schemes (Hammoudeh & Lowd, 2024). Retraining-based schemes evaluate attribution by repeatedly retraining LLM using different subsets, where representative approaches are leave-one-out (LOO) (Park et al., 2023), Shapley value (Jia et al., 2019; Wang & Jia, 2023), and datamodeling (Georgiev et al., 2024). However, these methods cannot be adopted for our purpose since the motivation for unlearning is to avoid retraining and increase efficiency. Gradient-based schemes assess the influence of training subsets on LLM generations by analyzing the training gradients. For example, influence-function-based methods (Koh & Liang, 2017; Han et al., 2020; Guo et al., 2021; Xia et al., 2024) approximate the LOO scores in different ways without retraining. However, they still incur large computations compared to our efficient approximation using a classifier, especially when invoked for a large number of test instances (Hammoudeh & Lowd, 2024).

Apart from these schemes, **text watermarking** has been utilized in recent works (Lau et al., 2024; Lu et al., 2025) for data attribution. By embedding verifiable watermarks into training data, text watermarking approaches enable computationally efficient data attribution at inference time. For example, Lau et al. (2024) adopts an LLM paraphraser to modify the semantics in a verifiable way, and Lu et al. (2025) directly embeds invisible Unicode characters into training texts. Despite the efficiency during verification, text watermarking tediously requires all training data to be watermarked before LLM training and unlearning, which makes it hard to be applied to LLMs that have already been trained without watermarks. Nonetheless, we compared our implementation of `DareU` to that using text watermarking in Sec. 5.3.

A parallel line of LLM attribution works focuses on **instance attribution** to identify the specific training sample that influences a piece of LLM-generated response. For example, many influence function-based methods (Kwon et al., 2024; Xia et al., 2024), Shapley-based methods (Wang et al., 2024; 2025a), and gradient tracing methods (Li et al., 2025) can effectively be used for instance attribution. While these works have been popular for research on interpretability and explainable AI, they are usually computationally expensive and are less relevant in our unlearning setting. Since we focus on identifying whether the forget set belonging to a group of forget owners (instead of individual data instances) still influences the generated responses, the high computational cost of instance-level attribution can be unnecessarily expensive.

## B. Unlearning from Fine-tuned or Pre-trained LLMs

In the main paper, we focus on unlearning from a LLM that is fine-tuned on a task-specific dataset. This setting is still significant as in many real-life applications, the practitioners lack resources to train an LLM from scratch and thus finetune a pre-trained LLM on their task-specific dataset to adapt to their task at hand instead. For example, a healthcare firm may download a pre-trained Llama2-7B model and then finetune it on private healthcare data. Such downstream users or practitioners may need to ensure privacy compliance and unlearn part of the private fine-tuning data as in popular unlearning benchmarks and methods (Maini et al., 2024; Shi et al., 2024b).

Unlearning pre-training data from pre-trained models is also an important real-world application. Such unlearning involves larger datasets/token numbers and new non-trivial computation challenges that specific papers, such as (Yao et al., 2024b) address. In practice, `DareU` can be applied to unlearning pre-trained models given both access to the pre-training dataset (or a subset of it) to train a data attribution classifier and sufficient compute for reinforcement learning. Initially, we excluded experiments on pre-trained models as (i) they were excluded by some unlearning papers (Zhang et al., 2024b) and (ii) it is too expensive to obtain the retrained pre-trained model for benchmarking/computing the ToW metric.

Our previous experiments were based on the efficient LoRA finetuning instead. As a proof of concept for pre-training, we applied DareU to full-parameter unlearning of the TOFU dataset for Llama2-7B, where the model was initially full-parameter

*Table 9.* Attribution scores and $\mathcal{D}_{\mathcal{F}}$ ROUGE measured on the baseline approaches and our `DareU`. The results demonstrate the correlation between attribution scores and ROUGE, showing that the de-attribution objective implies true unlearning.

| Approach | Attr. Score | $\mathcal{D}_{\mathcal{F}}$ ROUGE |
|---|---|---|
| Original | 0.738 ± 0.007 | 0.908 ± 0.007 |
| Fine-tune | 0.732 ± 0.009 | 0.772 ± 0.005 |
| GA | 0.013 ± 0.008 | 0.000 ± 0.000 |
| GDiff | 0.549 ± 0.099 | 0.419 ± 0.045 |
| SCRUB | 0.335 ± 0.183 | 0.268 ± 0.226 |
| SCRN | 0.743 ± 0.007 | 0.292 ± 0.135 |
| IDK | 0.394 ± 0.122 | 0.028 ± 0.003 |
| NPO | 0.164 ± 0.050 | 0.286 ± 0.018 |
| `DareU` | 0.324 ± 0.009 | 0.407 ± 0.015 |

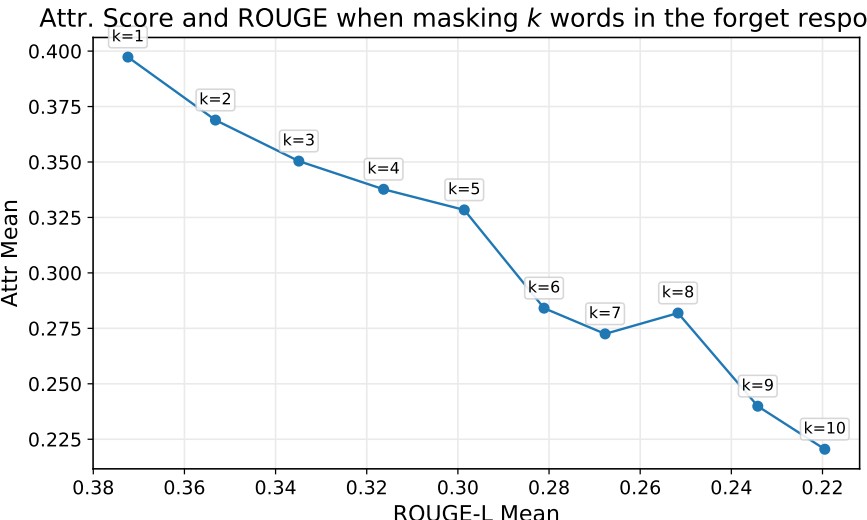

*Figure 4.* Illustration that the de-attribution objective correlates with true unlearning. By decreasing the attribution score through masking more words from the forget responses, the ROUGE score decreases as well.

trained on the TOFU dataset. Table 8 shows that DareU can still achieve a high ToW score.

## C. Data De-attribution Objective

Sec. 4.1 suggests that it is challenging to formally prove that the de-attribution objective implies true unlearning. Thus, we empirically demonstrate that lowering attribution scores better approximates the retrained LLM instead, under the condition that the attribution classifier accurately assigns high scores to responses influenced by $\mathcal{D}_{\mathcal{F}}$ and low scores to responses that are not (e.g., responses to queries about $\mathcal{D}_{\mathcal{R}}$ by the retrained LLM). In Tables 2 and 3, we have empirically observed that lower attribution scores and `DareU` lead to closer performance (e.g., ROUGE) to retrained LLM and yield better ToW. Table 9 further demonstrates the correlation between attribution scores and ROUGE.

Although there is neither a standard metric to measure true unlearning nor benchmark datasets or models with different levels of true unlearning, we can still verify if de-attribution correlates with true unlearning. We systematically construct different levels of true unlearning by masking the forget responses. Specifically, we mask $k$ words from the forget responses and measure the attribution score and ROUGE on the masked responses. Fig. 4 illustrates that as the attribution score decreases, the ROUGE score (which measures similarity of content and semantics) decreases as well. This suggests that the target forget content is truly unlearned, thereby validating that the de-attribution correlates well with true unlearning.

## D. Proximal Policy Optimization (PPO)

PPO (Schulman et al., 2017) is a widely adopted reward-based RL algorithm commonly employed for post-training LLMs. In this setting, the pre-trained LLM has been fine-tuned for a specific task, resulting in $\pi_{\mathcal{N}}$, and is further refined by using PPO to learn a policy (in our context, the LLM $\pi$). PPO utilizes an explicit reward model to compute the reward on a trajectory (in our context, the LLM-generated response $y$) and optimizes the expected reward while constraining the updated policy to not diverge too much from the old policy $\pi_{\text{old}}$ at the start of the epoch. Let $s_t(\theta) := \frac{\pi(y_t|c_t)}{\pi_{\text{old}}(y_t|c_t)}$ denote the importance sampling ratio between the new and old policies, where $\pi(y_t \mid c_t)$ denotes the probability of $\pi$ generating token $y_t$ given context $c_t$ and $\pi_{\text{old}}$ denotes the policy from the previous update. $\pi_{\text{old}}$ is used for the importance sampling ratio because the training data are collected under $\pi_{\text{old}}$. Importance sampling is required to evaluate the objective for the updated policy, which ensures that the estimate is unbiased under the new policy. The proof is shown in the following:

For any integrable function $f(c_t, y_t)$ conditioning on $c_t$, we have:

$$
\begin{aligned}
\mathbb{E}_{y_t \sim \pi_{\text{old}}(\cdot|c_t)}[s_t(\theta)\, f(c_t, y_t)] &= \sum_{y_t} \pi_{\text{old}}(y_t \mid c_t)\, \frac{\pi(y_t \mid c_t)}{\pi_{\text{old}}(y_t \mid c_t)}\, f(c_t, y_t) \\
&= \sum_{y_t} \pi(y_t \mid c_t)\, f(c_t, y_t) \\
&= \mathbb{E}_{y_t \sim \pi(\cdot|c_t)}[f(c_t, y_t)],
\end{aligned}
\tag{3}
$$

where the summation can be replaced by an integral for continuous $y_t$. In particular, taking $f(c_t, y_t) = \hat{A}_t$ yields

$$
\mathbb{E}_{y_t \sim \pi_{\text{old}}(\cdot|c_t)}\left[s_t(\theta)\, \hat{A}_t\right] = \mathbb{E}_{y_t \sim \pi(\cdot|c_t)}\left[\hat{A}_t\right],
\tag{4}
$$

assuming $\pi_{\text{old}}(y_t \mid c_t) > 0$ whenever $\pi(y_t \mid c_t) > 0$. Then, the advantage $\hat{A}_t$ for PPO update is defined as:

$$
\hat{A}_t = \hat{R}_t - V(c_t), \qquad \hat{R}_t = \sum_{j=0}^{T-t-1} \gamma^j\, r_{t+j},
\tag{5}
$$

where $\hat{R}_t$ denotes the empirical discounted return from step $t$ to the end of the trajectory, $V(\cdot)$ is a learned value-function baseline which can be used to reduces variance without introducing bias during training (Sutton et al., 1999), $r_t$ is the reward at step $t$, $\gamma \in (0, 1)$ is the discount factor, and $T$ is the trajectory length. The PPO clipped surrogate objective is then formulated as:

$$
\max_{\theta}\ \mathbb{E}_t\left[ \min\left(s_t(\theta)\hat{A}_t,\ \text{clip}(s_t(\theta), 1-\varepsilon, 1+\varepsilon)\hat{A}_t\right)\right],
\tag{6}
$$

where $\varepsilon > 0$ is a hyperparameter that controls the step size by clipping overly large policy updates, and the KL penalty to a fixed reference policy (in our context, the original LLM $\pi_{\mathcal{N}}$) is absorbed into the rewards used to compute $\hat{A}_t$. For a more comprehensive introduction of PPO, refer to Schulman et al. (2017).

While reward-free RL algorithms such as DPO (Rafailov et al., 2023) offer implementation simplicity, recent studies (Xu et al., 2024) have demonstrated that PPO achieves robust effectiveness and stability. On the other hand, some reward-based approaches like GRPO (Shao et al., 2024) require generating multiple responses for each query and are hence computationally more expensive. Overall, PPO offers a balanced trade-off between update stability and computational cost.

**Log-based Transformation.** In Sec. 4.2.1, we designed the log-based transformation for attribution reward as:

$$
\text{clip}(\ln(1 - \text{clip}(b, \varepsilon,\ 1-\varepsilon))\, /\, C_{\text{seg}},\ -1,\ 0).
$$

In our implementation, we set $C_{\text{seg}} = 1.05$ to assign a larger penalty to generations attributed to $\mathcal{D}_{\mathcal{F}}$. To elaborate, as shown in Fig. 5, the transformed attribution score $\text{DA}^{\chi}$ decreases rapidly to $-1$ when the raw attribution score is high. Since a higher attribution score indicates a higher probability that the generated response can be attributed to $\mathcal{D}_{\mathcal{F}}$, assigning a larger penalty allows `DareU` to quickly suppress these generations during unlearning. In contrast, when the raw attribution score is low, the penalty is moderate to better preserve the performance on the retain set.

## E. Detailed Experimental Setup

We conduct our experiments on one NVIDIA H200 GPU. The results are averaged over 3 random seeds 41, 42, and 43.

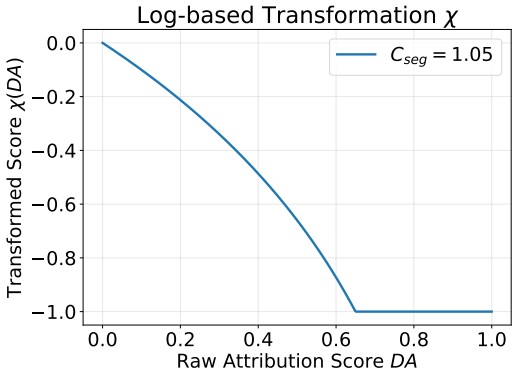

*Figure 5.* Effect of the log-based transformation. $C_{\text{seg}}$ controls the curvature of the log transform and the onset of clipping. Smaller values induce a sharp curvature, clip many inputs at $-1$, and reduce the reward/gradient informativeness. Larger values flatten the mapping and reward signals and potentially lead to insufficient penalization of high-score inputs.

### E.1. Training and Unlearning Configurations

**Fine-tuning Llama2-7b.** We fine-tune the pre-trained Llama-2-7B-chat model from Hugging Face[4] to obtain the original LLM $\pi_{\mathcal{N}}$ before unlearning. We use LoRA with $r = 8$, $\alpha = 16$, dropout $0.05$ for fine-tuning, with training batch size $16$ and learning rate $1.5 \times 10^{-4}$ for 5 epochs on TOFU and 15 epochs on ArXiv. We then fine-tune the TinyLlama-1.1B-Chat model[5] with a classification head to obtain the data attribution classifier. We use LoRA with $r = 16$, $\alpha = 32$, dropout $0.1$ for fine-tuning, with training batch size $16$ and learning rate $1 \times 10^{-4}$ for 5 epochs on TOFU and 15 epochs on ArXiv.

**Fine-tuning Qwen3-8b.** We fine-tune the pre-trained Qwen-8b model from Hugging Face[6] using to obtain the original LLM $\pi_{\mathcal{N}}$. We use LoRA with $r = 8$, $\alpha = 16$, dropout $0.05$ for fine-tuning, with training batch size $16$ and learning rate $1.5 \times 10^{-4}$ for 5 epochs on TOFU and 15 epochs on ArXiv. We fine-tune the Qwen/Qwen3-0.6B model[7] with a classification head to obtain the data attribution classifier. We use LoRA with $r = 16$, $\alpha = 32$, dropout $0.1$ for fine-tuning, with training batch size $16$ and learning rate $1 \times 10^{-4}$ for 5 epochs on TOFU and 15 epochs on ArXiv.

**Unlearning with `DareU`.** For `DareU`, we use data epoch 1, KL penalty coefficient $0.1$, value function coefficient $0.2$, PPO clip range $0.2$, and sampling temperature $0.8$ across all models and datasets. For other parameters, we use PPO update step $20$, train batch size $32$, and learning rate $1.5 \times 10^{-4}$ for both Llama2-7b and Qwen3-8b on TOFU; PPO update step $6$, train batch size $16$, and learning rate $2 \times 10^{-4}$ for Llama2-7b on Arxiv; PPO update step $8$, train batch size $8$, and learning rate $2 \times 10^{-4}$ for Qwen3-8b on Arxiv.

### E.2. Unlearning Baselines

We have considered the following baseline LLM unlearning approaches in our experiments. For a fair comparison, we have performed a search over the learning rates $1 \times 10^{-4}$, $1.5 \times 10^{-4}$, $3 \times 10^{-4}$, and $5 \times 10^{-4}$ for the best unlearning performance.

**Retraining:** fine-tune the pre-trained base LLM on $\mathcal{D}_{\mathcal{R}}$ using the same hyperparameters (described above in App. E.1) as those in fine-tuning the original LLM $\pi_{\mathcal{N}}$ before unlearning. Retraining produces the golden standard for other LLM unlearning approaches.

**Prediction Loss-based Approaches:**

- **Fine-tuning:** minimize the loss on $\mathcal{D}_{\mathcal{R}}$ by fine-tuning $\pi$ on $\mathcal{D}_{\mathcal{R}}$ for 1 epoch with the same hyperparameters during training. Fine-tuning assumes that the LLM naturally forgets $\mathcal{D}_{\mathcal{F}}$ with more training on $\mathcal{D}_{\mathcal{R}}$.

- **Gradient Ascent (GA):** maximize the loss on $\mathcal{D}_{\mathcal{F}}$ by fine-tuning $\pi$ on $\mathcal{D}_{\mathcal{F}}$ to revert its influence from training. We run

---

[4]https://huggingface.co/meta-llama/Llama-2-7B-chat-hf.
[5]https://huggingface.co/TinyLlama/TinyLlama-1.1B-Chat-v1.0.
[6]https://huggingface.co/Qwen/Qwen3-8B.
[7]https://huggingface.co/Qwen/Qwen3-0.6B.

*Table 10.* Unlearning performance of `DareU` when implemented with different reward designs on Llama2-7B × TOFU.

| Approach | $\mathcal{D}_{\mathcal{F}}$ ROUGE ($\rightarrow$) | $\mathcal{D}_{\mathcal{R}}$ ROUGE ($\rightarrow$) | $\mathcal{D}_{test}$ ROUGE ($\rightarrow$) | ToW ($\uparrow$) | Truth Ratio ($\uparrow$) | MIA ($\rightarrow$) |
|---|---|---|---|---|---|---|
| Original | 0.908 ± 0.007 | 0.901 ± 0.003 | 0.812 ± 0.014 | 0.455 ± 0.010 | 0.508 ± 0.001 | 1.0000 ± 0.0000 |
| Retraining | 0.381 ± 0.004 | 0.917 ± 0.009 | 0.818 ± 0.030 | 1.000 ± 0.000 | 0.669 ± 0.002 | 0.9971 ± 0.0025 |
| Sequence-level reward | 0.631 ± 0.009 | **0.813 ± 0.016** | 0.793 ± 0.007 | 0.655 ± 0.012 | 0.545 ± 0.004 | 0.9940 ± 0.0056 |
| Per-token reward | 0.427 ± 0.009 | 0.804 ± 0.007 | 0.797 ± 0.015 | **0.822 ± 0.015** | 0.561 ± 0.003 | 0.9992 ± 0.0006 |
| Ours ($\kappa$ token reward) | **0.407 ± 0.015** | 0.797 ± 0.006 | **0.834 ± 0.030** | 0.816 ± 0.011 | **0.564 ± 0.005** | **0.9962 ± 0.0042** |

*Table 11.* Unlearning performance of `DareU` for different sizes of $\mathcal{D}_{\mathcal{F}}$ on Llama2 × TOFU.

| Size of $\mathcal{D}_{\mathcal{F}}$ | Approach | $\mathcal{D}_{\mathcal{F}}$ ROUGE ($\rightarrow$) | $\mathcal{D}_{\mathcal{R}}$ ROUGE ($\rightarrow$) | $\mathcal{D}_{test}$ ROUGE ($\rightarrow$) | ToW ($\uparrow$) | MIA ($\rightarrow$) |
|---|---|---|---|---|---|---|
| 0 | Original | 0.908 ± 0.007 | 0.901 ± 0.003 | 0.812 ± 0.014 | 0.455 ± 0.010 | 1.0000 ± 0.0000 |
| 200 | Retraining | 0.379 ± 0.004 | 0.911 ± 0.006 | 0.830 ± 0.009 | 1.000 ± 0.000 | 0.9999 ± 0.0001 |
| | `DareU` | 0.386 ± 0.019 | 0.803 ± 0.001 | 0.796 ± 0.038 | 0.826 ± 0.035 | 1.0000 ± 0.0000 |
| 400 | Retraining | 0.381 ± 0.004 | 0.917 ± 0.009 | 0.818 ± 0.030 | 1.000 ± 0.000 | 0.9971 ± 0.0025 |
| | `DareU` | 0.407 ± 0.015 | 0.797 ± 0.006 | 0.834 ± 0.030 | 0.816 ± 0.011 | 0.9962 ± 0.0042 |
| 800 | Retraining | 0.371 ± 0.005 | 0.897 ± 0.011 | 0.812 ± 0.014 | 1.000 ± 0.000 | 0.9998 ± 0.0002 |
| | `DareU` | 0.463 ± 0.014 | 0.811 ± 0.007 | 0.797 ± 0.009 | 0.812 ± 0.010 | 0.9984 ± 0.0018 |

GA with learning rate $1.5 \times 10^{-4}$ on TOFU and $1 \times 10^{-4}$ on ArXiv for 1 epoch.

- **Gradient Difference** (GDiff) (Maini et al., 2024): optimize the weighted average of the loss on $\mathcal{D}_{\mathcal{R}}$ and the negated loss on $\mathcal{D}_{\mathcal{F}}$. We sample $\mathcal{D}_{\mathcal{R}}$ to match the size of $\mathcal{D}_{\mathcal{F}}$ and run GDiff with learning rate $1.5 \times 10^{-4}$ on TOFU and $1 \times 10^{-4}$ on ArXiv for 1 epoch.

- **SCRUB** (Kurmanji et al., 2023): maximize the KL divergence from the output distribution of the original LLM on $\mathcal{D}_{\mathcal{F}}$ while minimizing that on $\mathcal{D}_{\mathcal{R}}$ and minimizing the task loss on $\mathcal{D}_{\mathcal{R}}$. We run SCRUB with learning rate $1.5 \times 10^{-4}$ on TOFU and $3 \times 10^{-4}$ on ArXiv for 3 epochs.

- **SCuReNewton** (SCRN) (Bui et al., 2026): leverages a cubic-regularized Newton's update as a second-order approach for LLM unlearning. We set $M = 400$ and run 2 outer steps and 4 inner steps.

**Preference Optimization Methods:**

- **IDK** (Maini et al., 2024): encourage responses with "I don't know" for queries from $\mathcal{D}_{\mathcal{F}}$ using direct preference optimization. Note that since the refusal response for ArXiv is unclear, we only run IDK on TOFU. We run IDK with learning rate $1.5 \times 10^{-4}$ for 1 epoch.

- **Negative Preference Optimization (NPO)** (Zhang et al., 2024b): performs preference optimization with only $\mathcal{D}_{\mathcal{F}}$ samples as negative responses to discourage the original responses on $\mathcal{D}_{\mathcal{F}}$. We run NPO with learning rate $1.5 \times 10^{-4}$ on TOFU and $1 \times 10^{-4}$ on ArXiv for 3 epochs.

# F. Additional Experiments

### F.1. Reward Design Choice

As discussed in Sec. 4.2.1, in `DareU` we design an efficient computation of token-level reward $r_{t,d}$. While directly computing the reward for the entire sequence of $d$ can be more efficient, a sequence-level reward can be too sparse to consistently differentiate retain and forget behaviors, e.g., the difference between retain and forget tokens. To validate the effectiveness of our design ($\kappa$ token reward where $\kappa > 1$), we implement `DareU` with a sequence-level reward (i.e., directly computing $r_d$ without considering $t$ at all) and a per-token reward (i.e., when $\kappa = 1$) and compare their performance in Table 10. The results demonstrate that `DareU`, with our $\kappa$ token reward computation design, significantly outperforms the sequence-level reward and achieves comparable results to per-token reward. This validates the effectiveness of our reward computation design.

*Table 12.* Sensitivity of `DareU` to different learning rates on Llama2 × TOFU.

| Learning Rate | $\mathcal{D}_{\mathcal{F}}$ ROUGE ($\rightarrow$) | $\mathcal{D}_{\mathcal{R}}$ ROUGE ($\rightarrow$) | $\mathcal{D}_{test}$ ROUGE ($\rightarrow$) | ToW ($\uparrow$) | Truth Ratio ($\uparrow$) | MIA ($\rightarrow$) |
|---|---|---|---|---|---|---|
| Retraining | $0.381 \pm 0.004$ | $0.917 \pm 0.009$ | $0.818 \pm 0.030$ | $1.000 \pm 0.000$ | $0.669 \pm 0.002$ | $0.9971 \pm 0.0025$ |
| $1 \times 10^{-4}$ | $0.470 \pm 0.004$ | $0.822 \pm 0.003$ | $0.801 \pm 0.025$ | $0.792 \pm 0.021$ | $0.548 \pm 0.006$ | $1.0000 \pm 0.0000$ |
| $1.5 \times 10^{-4}$ | *$0.407 \pm 0.015$* | *$0.797 \pm 0.006$* | *$0.834 \pm 0.030$* | *$0.816 \pm 0.011$* | *$0.564 \pm 0.005$* | *$0.9962 \pm 0.0042$* |
| $2 \times 10^{-4}$ | $0.366 \pm 0.013$ | $0.761 \pm 0.005$ | $0.775 \pm 0.048$ | $0.781 \pm 0.035$ | $0.558 \pm 0.006$ | $0.9807 \pm 0.0128$ |

*Table 13.* Sensitivity of `DareU` to different numbers of data epochs on Llama2 × TOFU.

| Epoch | $\mathcal{D}_{\mathcal{F}}$ ROUGE ($\rightarrow$) | $\mathcal{D}_{\mathcal{R}}$ ROUGE ($\rightarrow$) | $\mathcal{D}_{test}$ ROUGE ($\rightarrow$) | ToW ($\uparrow$) | Truth Ratio ($\uparrow$) | MIA ($\rightarrow$) |
|---|---|---|---|---|---|---|
| Retraining | $0.381 \pm 0.004$ | $0.917 \pm 0.009$ | $0.818 \pm 0.030$ | $1.000 \pm 0.000$ | $0.669 \pm 0.002$ | $0.9971 \pm 0.0025$ |
| *1* | *$0.407 \pm 0.015$* | *$0.797 \pm 0.006$* | *$0.834 \pm 0.030$* | *$0.816 \pm 0.011$* | *$0.564 \pm 0.005$* | *$0.9962 \pm 0.0042$* |
| 2 | $0.323 \pm 0.010$ | $0.786 \pm 0.015$ | $0.785 \pm 0.022$ | $0.793 \pm 0.035$ | $0.590 \pm 0.009$ | $0.9987 \pm 0.0011$ |
| 3 | $0.274 \pm 0.026$ | $0.784 \pm 0.002$ | $0.794 \pm 0.008$ | $0.749 \pm 0.028$ | $0.581 \pm 0.008$ | $0.9984 \pm 0.0015$ |

## F.2. Scalability

In the main experiments, we have considered the unlearning of $10\%$ of the full TOFU dataset as $\mathcal{D}_{\mathcal{F}}$. To evaluate the scalability of `DareU` for different sizes of $\mathcal{D}_{\mathcal{F}}$, we further consider $5\%$ (200 samples) and $20\%$ (800 samples) of TOFU as $\mathcal{D}_{\mathcal{F}}$. For this experiment, Truth Ratio from (Maini et al., 2024) cannot be computed due to the lack of a perturbed set for $\mathcal{D}_{\mathcal{F}}$ of 800 samples. Experimental results in Table 11 demonstrate that `DareU` consistently achieves effective unlearning with high ToW scores across different sizes of $\mathcal{D}_{\mathcal{F}}$, which confirms the scalability of `DareU` to unlearn $\mathcal{D}_{\mathcal{F}}$ of different sizes.

## F.3. Sensitivity to Hyperparameters

In this section, we analyze the impact of different hyperparameters on the unlearning performance of our `DareU`. While PPO inherently involves several hyperparameters, we empirically validate that `DareU` is not sensitive to the change of hyperparameters within a reasonable range.

**Learning Rate.** Table 12 shows the unlearning performance of `DareU` with different learning rates using Llama2 on the TOFU dataset. The results show that `DareU` achieves consistent unlearning performance with high ToW scores across different learning rates. This validates the robustness of our `DareU` to the choice of learning rates.

**Data Epochs.** In our implementation of Algo. 1, we used data epoch $= 1$ and loop through $\mathcal{D}_{\mathcal{N}}$ once. In practice, given more computational resources, it might be possible to increase the number of data epochs. As demonstrated in Table 13, `DareU` unlearns more on $\mathcal{D}_{\mathcal{F}}$ as the number of data epochs increases. Note that, as we adopt the attribution classifier as an imperfect approximation of the attribution function, it may produce imperfect predictions even for samples that are already unlearned, which could risk over-forgetting. Nonetheless, `DareU` still outperforms many prediction loss-based approaches like GA and GDiff.

**PPO Update Steps.** Here, we evaluate the effect of adjusting the number of PPO update steps $S_{\text{sample}}$, which controls the number of PPO updates per sampled batch. As observed from the results in Table 14, `DareU` consistently achieves effective unlearning with different numbers of PPO epochs.

**Length of Tokens for Reward Computation.** As discussed in Sec. 4.2.1, we reduce the reward computational costs by dividing the sequence into slices of length $\kappa$. In practice, instead of directly modifying $\kappa$, we vary the number of slices $k = \frac{T}{\kappa}$ such that as $k$ increases towards $T$, our approach reduces to per-token reward computation in App. F.1. Table 15 validates that a larger $k$ (implying a smaller $\kappa$) yields better unlearning performance at the cost of increased computation.

**Distillation Regularization Factor.** Here, we analyze the impact of the distillation regularization factor $\lambda_{\text{dis}}$ introduced in Sec. 4.2.1. As demonstrated in Table 16, a larger $\lambda_{\text{dis}}$ imposes a stronger constraint and preserves more model utility. Nonetheless, `DareU` consistently maintains effective unlearning performance across different values of $\lambda_{\text{dis}}$.

Overall, these experiments show that `DareU` remains effective with different hyperparameters, which confirms that `DareU` is not sensitive to the change in hyperparameters within reasonable ranges. This validates the stability of our `DareU` and

*Table 14.* Sensitivity of `DareU` to different numbers of PPO update steps on Llama2 × TOFU.

| Step | $\mathcal{D}_\mathcal{F}$ ROUGE ($\rightarrow$) | $\mathcal{D}_\mathcal{R}$ ROUGE ($\rightarrow$) | $\mathcal{D}_{test}$ ROUGE ($\rightarrow$) | ToW ($\uparrow$) | Truth Ratio ($\uparrow$) | MIA ($\rightarrow$) |
|---|---|---|---|---|---|---|
| Retraining | 0.381 ± 0.004 | 0.917 ± 0.009 | 0.818 ± 0.030 | 1.000 ± 0.000 | 0.669 ± 0.002 | 0.9971 ± 0.0025 |
| 15 | 0.468 ± 0.012 | 0.806 ± 0.008 | 0.791 ± 0.013 | 0.791 ± 0.013 | 0.554 ± 0.004 | 0.9996 ± 0.0004 |
| *20* | *0.407 ± 0.015* | *0.797 ± 0.006* | *0.834 ± 0.030* | *0.816 ± 0.011* | *0.564 ± 0.005* | *0.9962 ± 0.0042* |
| 25 | 0.392 ± 0.007 | 0.786 ± 0.012 | 0.786 ± 0.010 | 0.832 ± 0.022 | 0.562 ± 0.003 | 0.9988 ± 0.0011 |

*Table 15.* Sensitivity of `DareU` to different numbers of slices $k$ on Llama2 × TOFU.

| $k$ Slices | $\mathcal{D}_\mathcal{F}$ ROUGE ($\rightarrow$) | $\mathcal{D}_\mathcal{R}$ ROUGE ($\rightarrow$) | $\mathcal{D}_{test}$ ROUGE ($\rightarrow$) | ToW ($\uparrow$) | Truth Ratio ($\uparrow$) | MIA ($\rightarrow$) |
|---|---|---|---|---|---|---|
| Retraining | 0.381 ± 0.004 | 0.917 ± 0.009 | 0.818 ± 0.030 | 1.000 ± 0.000 | 0.669 ± 0.002 | 0.9971 ± 0.0025 |
| 10 | 0.427 ± 0.006 | 0.803 ± 0.008 | 0.804 ± 0.021 | 0.810 ± 0.025 | 0.562 ± 0.005 | 0.9994 ± 0.0007 |
| *15* | *0.407 ± 0.015* | *0.797 ± 0.006* | *0.834 ± 0.030* | *0.816 ± 0.011* | *0.564 ± 0.005* | *0.9962 ± 0.0042* |
| 20 | 0.390 ± 0.004 | 0.799 ± 0.009 | 0.837 ± 0.018 | 0.825 ± 0.020 | 0.563 ± 0.005 | 0.9998 ± 0.0001 |

*Table 16.* Sensitivity of `DareU` to different $\lambda_{dis}$ on Llama2 × TOFU.

| $\lambda_{dis}$ | $\mathcal{D}_\mathcal{F}$ ROUGE ($\rightarrow$) | $\mathcal{D}_\mathcal{R}$ ROUGE ($\rightarrow$) | $\mathcal{D}_{test}$ ROUGE ($\rightarrow$) | ToW ($\uparrow$) | Truth Ratio ($\uparrow$) | MIA ($\rightarrow$) |
|---|---|---|---|---|---|---|
| Retraining | 0.381 ± 0.004 | 0.917 ± 0.009 | 0.818 ± 0.030 | 1.000 ± 0.000 | 0.669 ± 0.002 | 0.9971 ± 0.0025 |
| 1 | 0.385 ± 0.018 | 0.767 ± 0.010 | 0.780 ± 0.006 | 0.805 ± 0.008 | 0.561 ± 0.002 | 0.9954 ± 0.0032 |
| *2* | *0.407 ± 0.015* | *0.797 ± 0.006* | *0.834 ± 0.030* | *0.816 ± 0.011* | *0.564 ± 0.005* | *0.9962 ± 0.0042* |
| 3 | 0.441 ± 0.004 | 0.814 ± 0.009 | 0.789 ± 0.042 | 0.794 ± 0.036 | 0.561 ± 0.004 | 0.9865 ± 0.0154 |

suggests that `DareU` can be deployed with minimal hyperparameter tuning.

### F.4. Sequential Unlearning

In the setting of sequential unlearning, multiple subsets of $\mathcal{D}_\mathcal{F}$ are unlearned sequentially. This scenario is relevant when unlearning requests from data owners arrive sequentially or when computational resource constraints limit the size of the subset that can be unlearned concurrently. Sequential unlearning presents a more challenging unlearning task than batch unlearning in our main experiments because minor errors from an unlearning approach that may be negligible in a single unlearning round could accumulate through multiple rounds and lead to degraded unlearning performance. In this experiment, we consider unlearning 80 samples from $\mathcal{D}_\mathcal{F}$ of the TOFU dataset in each round, such that the full $\mathcal{D}_\mathcal{F}$ with a total of 400 samples is unlearned in 5 rounds. As visualized in Fig. 6, `DareU` consistently achieves $\mathcal{D}_\mathcal{F}$ ROUGE close to that of Retraining while preserving a decent $\mathcal{D}_\mathcal{R}$ ROUGE across 5 unlearning rounds.

### F.5. Unlearning Robustness

#### F.5.1. ROBUSTNESS TO ATTRIBUTION INACCURACIES

In our main experiments, we have considered different attribution classifiers for `DareU`, which generally have good accuracy (around 80%) and consistently lead to better unlearning performance than other unlearning methods (Table 6 vs. Table 2). Here, we evaluate the robustness of `DareU` when the attribution classifiers are inaccurate. In this experiment, we add noise to the classifiers and artificially reduce their accuracy. As shown in Table 18, the ToW score tends to drop with lower accuracy, but `DareU` still achieves a relatively high ToW across classifiers with different accuracies. This suggests that `DareU` is robust to inaccuracies in the attribution classifier.

#### F.5.2. UNLEARNING SEMANTIC OVERLAPS

A potential risk of using data de-attribution as the LLM unlearning objective is the over-reliance on semantics in the generated response. For example, if the attribution function merely relies on the semantics, it may fail to distinguish between the forget samples and semantically similar retain samples, hence influencing the unlearning performance. In this section, we investigate the robustness of `DareU` to such semantic overlaps. Specifically, we additionally incorporate 400

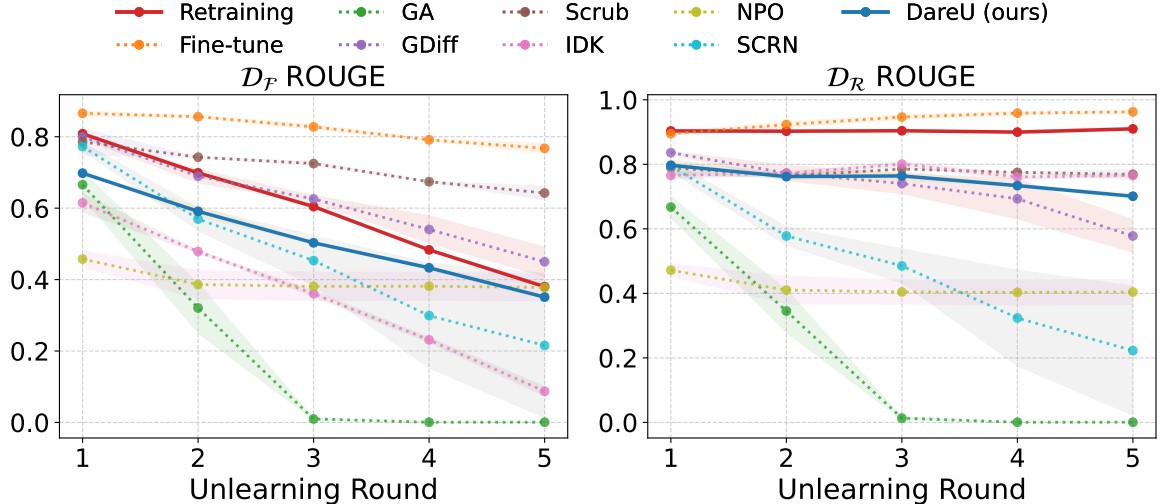

*Figure 6.* Unlearning performance over 5 unlearning rounds under the sequential unlearning setting on Llama2 × TOFU.

*Table 17.* Unlearning performance on Llama2 × TOFU when there are semantic overlaps in $\mathcal{D}_\mathcal{F}$ and $\mathcal{D}_\mathcal{R}$.

| Approach | $\mathcal{D}_\mathcal{F}$ ROUGE ($\rightarrow$) | $\mathcal{D}_\mathcal{R}$ ROUGE ($\rightarrow$) | $\mathcal{D}_{test}$ ROUGE ($\rightarrow$) | ToW ($\uparrow$) | MIA ($\rightarrow$) |
|---|---|---|---|---|---|
| Original | 0.768 ± 0.010 | 0.880 ± 0.006 | 0.808 ± 0.028 | 0.745 ± 0.028 | 1.0000 ± 0.0001 |
| Retraining | 0.559 ± 0.003 | 0.905 ± 0.005 | 0.774 ± 0.019 | 1.000 ± 0.000 | 0.9966 ± 0.0047 |
| Fine-tune | **0.599 ± 0.005** | **0.879 ± 0.003** | **0.769 ± 0.018** | **0.916 ± 0.013** | 0.9999 ± 0.0001 |
| GA | 0.001 ± 0.000 | 0.001 ± 0.000 | 0.000 ± 0.000 | 0.009 ± 0.001 | 0.9989 ± 0.0006 |
| GDiff | 0.003 ± 0.001 | 0.101 ± 0.046 | 0.066 ± 0.030 | 0.025 ± 0.004 | 0.9998 ± 0.0002 |
| SCRUB | 0.406 ± 0.255 | 0.426 ± 0.273 | 0.535 ± 0.379 | 0.450 ± 0.306 | 0.8341 ± 0.2341 |
| SCRN | 0.001 ± 0.000 | 0.001 ± 0.000 | 0.000 ± 0.000 | 0.009 ± 0.001 | 1.0000 ± 0.0001 |
| IDK | 0.039 ± 0.010 | 0.797 ± 0.015 | 0.763 ± 0.028 | 0.421 ± 0.010 | 1.0000 ± 0.0000 |
| NPO | 0.329 ± 0.019 | 0.320 ± 0.018 | 0.467 ± 0.039 | 0.221 ± 0.012 | 0.9996 ± 0.0006 |
| DareU | 0.453 ± 0.007 | 0.798 ± 0.008 | 0.805 ± 0.009 | 0.797 ± 0.010 | **0.9888 ± 0.0149** |

paraphrased samples from $\mathcal{D}_\mathcal{F}$ into $\mathcal{D}_\mathcal{R}$. Ideally, a robust unlearning approach should still unlearn $\mathcal{D}_\mathcal{F}$ without influencing the performance on $\mathcal{D}_\mathcal{R}$ even with these semantic overlaps. The results in Table 17 demonstrate that the unlearning performance of `DareU` remains unaffected with a high ToW score despite the presence of semantic overlaps. This validates the robustness of `DareU` to perform unlearning when there is semantically similar data.

### F.5.3. ADVERSARIAL PROBING

In this work, we employ a lightweight classifier to approximate the ideal attribution function. While efficient, there is a risk that the unlearned LLM bypasses the classifier by learning a policy that produces low attribution scores merely for specific queries without genuinely unlearning the content from $\mathcal{D}_\mathcal{F}$. To verify genuine unlearning, we introduce adversarial probing using paraphrased queries from $\mathcal{D}_\mathcal{F}$. If `DareU` merely overfits to the specific forget queries to bypass the de-attribution objective without genuinely unlearning the content from $\mathcal{D}_\mathcal{F}$, it would still be able to generate responses to paraphrased queries, producing higher ROUGE scores closer to the original LLM. Experimental results in Table 19 demonstrate that `DareU` achieves a $\mathcal{D}_\mathcal{F}$ ROUGE score close to Retraining while maintaining a high ToW$_{para}$ score even with paraphrased forget queries. This validates the robustness of our `DareU` against adversarial probing.

### F.5.4. RELEARNING

We evaluate whether `DareU` and the baseline unlearning approaches prevent the unlearned LLM from rapidly recovering the unlearned knowledge through subsequent fine-tuning, following the "relearning" setting in WDMP (Li et al., 2024). Specifically, we first perform normal unlearning on Llama2 × TOFU and subsequently fine-tune the unlearned LLMs on $\mathcal{D}_\mathcal{F}$. We visualize the gain in $\mathcal{D}_\mathcal{F}$ ROUGE in Fig. 7, which is normalized to 0 at the start of the relearning process. The results indicate that while `DareU` naturally falls short of Retraining, it exhibits stronger resistance than other baseline approaches, such as GA, NPO, and IDK, where the unlearned LLM rapidly recovers the unlearned knowledge.

*Table 18.* Unlearning performance on inaccuracy attribution classifiers. While the ToW score inevitably drops with inaccurate attribution classifiers, `DareU` still preserves a relatively high ToW.

| Classifier Acc. | $\mathcal{D}_{\mathcal{F}}$ ROUGE ($\rightarrow$) | $\mathcal{D}_{\mathcal{R}}$ ROUGE ($\rightarrow$) | $\mathcal{D}_{test}$ ROUGE ($\rightarrow$) | ToW ($\uparrow$) | MIA ($\rightarrow$) |
|---|---|---|---|---|---|
| *0.877* | *0.407 ± 0.015* | *0.797 ± 0.006* | *0.834 ± 0.030* | *0.816 ± 0.011* | *0.9888 ± 0.0042* |
| 0.814 | 0.413 ± 0.008 | 0.786 ± 0.008 | 0.823 ± 0.022 | 0.802 ± 0.014 | 0.9870 ± 0.0002 |
| 0.760 | 0.425 ± 0.013 | 0.791 ± 0.010 | 0.828 ± 0.025 | 0.800 ± 0.016 | 0.9845 ± 0.0011 |
| 0.678 | 0.413 ± 0.005 | 0.783 ± 0.007 | 0.820 ± 0.021 | 0.795 ± 0.015 | 0.9808 ± 0.0037 |
| 0.488 | 0.440 ± 0.006 | 0.769 ± 0.007 | 0.806 ± 0.021 | 0.771 ± 0.012 | 0.9735 ± 0.0032 |

*Table 19.* Querying the unlearned LLM with paraphrased questions based on the training data. Results are measured on Llama2 $\times$ TOFU. We use ToW$_{\text{para}}$ to denote the ToW score computed with paraphrased $\mathcal{D}_{\mathcal{F}}$ ROUGE.

| Approach | $\mathcal{D}_{\mathcal{F}}$ ROUGE ($\rightarrow$) | paraphrased $\mathcal{D}_{\mathcal{F}}$ ROUGE ($\rightarrow$) | $\mathcal{D}_{\mathcal{R}}$ ROUGE ($\rightarrow$) | $\mathcal{D}_{test}$ ROUGE ($\rightarrow$) | ToW ($\uparrow$) | ToW$_{\text{para}}$ ($\uparrow$) |
|---|---|---|---|---|---|---|
| Original | 0.908 ± 0.007 | 0.522 ± 0.008 | 0.901 ± 0.003 | 0.812 ± 0.014 | 0.455 ± 0.010 | 0.792 ± 0.013 |
| Retraining | 0.381 ± 0.004 | 0.348 ± 0.005 | 0.917 ± 0.009 | 0.818 ± 0.030 | 1.000 ± 0.000 | 1.000 ± 0.000 |
| Fine-tune | 0.772 ± 0.005 | 0.466 ± 0.007 | **0.887 ± 0.004** | 0.775 ± 0.013 | 0.563 ± 0.017 | 0.834 ± 0.014 |
| GA | 0.000 ± 0.000 | 0.000 ± 0.000 | 0.001 ± 0.001 | 0.000 ± 0.000 | 0.009 ± 0.001 | 0.010 ± 0.002 |
| GDiff | 0.419 ± 0.045 | **0.334 ± 0.033** | 0.601 ± 0.074 | 0.760 ± 0.026 | 0.614 ± 0.057 | 0.615 ± 0.073 |
| SCRUB | 0.268 ± 0.226 | 0.209 ± 0.148 | 0.302 ± 0.266 | 0.283 ± 0.364 | 0.219 ± 0.265 | 0.253 ± 0.312 |
| SCRN | 0.292 ± 0.135 | 0.277 ± 0.123 | 0.270 ± 0.115 | 0.657 ± 0.461 | 0.251 ± 0.161 | 0.253 ± 0.159 |
| IDK | 0.028 ± 0.003 | 0.033 ± 0.002 | 0.804 ± 0.012 | 0.795 ± 0.026 | 0.548 ± 0.007 | 0.580 ± 0.012 |
| NPO | 0.286 ± 0.018 | 0.230 ± 0.029 | 0.318 ± 0.011 | 0.473 ± 0.102 | 0.239 ± 0.047 | 0.171 ± 0.034 |
| `DareU` | **0.407 ± 0.015** | 0.321 ± 0.011 | 0.797 ± 0.006 | **0.834 ± 0.030** | **0.816 ± 0.011** | **0.837 ± 0.031** |

## F.6. Generation Collapse Evaluation

In Sec. 3, we motivated the data de-attribution objective as it does not favor gibberish responses. Here, we evaluate whether the unlearned LLMs from both our `DareU` and the baseline unlearning approaches produce gibberish outputs. In Table 20, we report the percentage of gibberish responses generated across all forget queries. Qualitative examples of the gibberish are shown in Figure 8. The results show that DareU produces less gibberish than most prediction loss-based unlearning approaches. This may be due to our design of the distillation regularization term (Sec. 4.2). Note that Fine-tune does not effectively unlearn and IDK makes the unlearned model outputs "I don't know", hence producing 0 percent gibberish in the generated outputs.

## G. Time Complexity per PPO Update

**Notation.** Let $B$ be the local batch size per update, $Q$ the query length, $T$ the generated length, and $L = Q + T$ the total sequence length. Let $S_{\text{PPO}}$ be the number of PPO update steps, $m$ the micro-batch size, and $k$ the number of prefix slices from our reward computation. We denote vocabulary size by $v$.

We abstract model costs as: policy forward/backward ($F_\pi(L)$, $B_\pi(L)$), value forward/backward ($F_{\text{val}}(L)$, $B_{\text{val}}(L)$), reference forward $F_{\text{ref}}(L)$, and classifier reward forward $F_{\text{cls}}(L)$.

**Rollout cost excluding reward.** The rollout stage includes autoregressive generation, policy/ref log-prob computations, value prediction, and token-level post-processing like masking, KL, whitening, GAE (lines 13-14 and partially line 15 in Algo. 1). We approximate the non-reward rollout complexity as:

$$C_{\neg r} \approx \underbrace{\Theta(BT\,F_\pi(L))}_{\text{Sequential Generation}} + \underbrace{\Theta(BF_{\text{ref}}(L)) + \Theta(BF_{\text{val}}(L))}_{\text{Forward Pass}} + \Theta(BTv), \tag{7}$$

where the $BTv$ term accounts for vocabulary-normalized operations like log-softmax and entropy computation, and $BT$ denotes the total number of generated tokens processed in one PPO update.

**Reward computation.** Our reward is computed by evaluating a data classifier on $k$ increasing prefixes per sample Thus for per update reward, the cost is:

$$C_r := \Theta(Bk\,F_{\text{cls}}(L)) + \Theta(BkL). \tag{8}$$

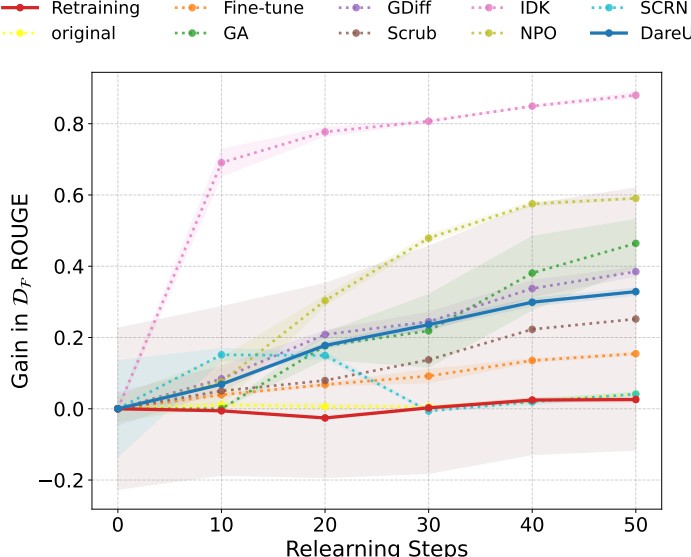

*Figure 7.* Relearning result where the unlearned LLM is fine-tuned on $\mathcal{D}_{\mathcal{F}}$ on Llama2 $\times$ TOFU.

*Table 20.* The percentage of collapsed generations after unlearning.

| Approach | $\mathcal{D}_{\mathcal{F}}$ ROUGE ($\rightarrow$) | paraphrased $\mathcal{D}_{\mathcal{F}}$ ROUGE ($\rightarrow$) | $\mathcal{D}_{\mathcal{R}}$ ROUGE ($\rightarrow$) | $\mathcal{D}_{test}$ ROUGE ($\rightarrow$) | ToW ($\uparrow$) | pct. gibberish ($\downarrow$) |
|---|---|---|---|---|---|---|
| Original | 0.908 ± 0.007 | 0.522 ± 0.008 | 0.901 ± 0.003 | 0.812 ± 0.014 | 0.455 ± 0.010 | 0.00 ± 0.00 |
| Retraining | 0.381 ± 0.004 | 0.348 ± 0.005 | 0.917 ± 0.009 | 0.818 ± 0.030 | 1.000 ± 0.000 | 0.00 ± 0.00 |
| Fine-tune | 0.772 ± 0.005 | 0.466 ± 0.007 | **0.887 ± 0.004** | 0.775 ± 0.013 | 0.563 ± 0.017 | 0.00 ± 0.00 |
| GA | 0.000 ± 0.000 | 0.000 ± 0.000 | 0.001 ± 0.001 | 0.000 ± 0.000 | 0.009 ± 0.001 | 100.00 ± 0.00 |
| GDiff | 0.419 ± 0.045 | **0.334 ± 0.033** | 0.601 ± 0.074 | 0.760 ± 0.026 | 0.614 ± 0.057 | 0.00 ± 0.00 |
| SCRUB | 0.268 ± 0.226 | 0.209 ± 0.148 | 0.302 ± 0.266 | 0.283 ± 0.364 | 0.219 ± 0.265 | 45.25 ± 0.00 |
| SCRN | 0.292 ± 0.135 | 0.277 ± 0.123 | 0.270 ± 0.115 | 0.657 ± 0.461 | 0.251 ± 0.161 | 10.75 ± 0.00 |
| IDK | 0.028 ± 0.003 | 0.033 ± 0.002 | 0.804 ± 0.012 | 0.795 ± 0.026 | 0.548 ± 0.007 | 0.00 ± 0.00 |
| NPO | 0.286 ± 0.018 | 0.230 ± 0.029 | 0.318 ± 0.011 | 0.473 ± 0.102 | 0.239 ± 0.047 | 23.58 ± 0.00 |
| DareU | **0.407 ± 0.015** | 0.321 ± 0.011 | 0.797 ± 0.006 | **0.834 ± 0.030** | **0.816 ± 0.011** | 4.08 ± 0.00 |

The first term is the forward cost for the classifier when computing the attribution reward. The second term represents the complexity to construct the prefixes. The first term dominates in practice, while the second term accounts for prefix construction and padding overhead. Using a learned reward model/classifier to score generated sequences is standard in RLHF style fine-tuning (Ziegler et al., 2019).

**Rollout cost in total**   For clarity, the full rollout cost per update is:

$$C_{\text{roll}} := C_{\neg r} + C_r. \tag{9}$$

**PPO optimization.**   PPO update performs $S_{\text{PPO}}$ steps; each update step processes all $B$ samples in micro-batches of size $m$, hence $B/m$ optimization steps per update step. We define the average per-step cost as:

$$C_{\text{step}} := \Theta(F_\pi(L) + B_\pi(L)) + \Theta(F_{\text{val}}(L) + B_{\text{val}}(L)) + \Theta(\rho_{distill} F_{\text{ref}}(L)) + \Theta(Tv), \tag{10}$$

where $\rho_{distill} \in [0, 1]$ represents the distillation process during PPO update, and $Tv$ again accounts for vocabulary-scale normalization. Therefore, the optimization cost per update is

$$C_{\text{opt}} := \Theta\Big(S_{\text{PPO}} \cdot \frac{B}{m} \cdot C_{\text{step}}\Big). \tag{11}$$

**Total cost per update.**   Combining rollout complexity and optimization complexity, the whole cost yields

$$C_{\text{update}} = C_{\text{roll}} + C_{\text{opt}}. \tag{12}$$

**Typical GA Gibberish Example**

[Gibberish: Severely Repeated Tokens]

en en en en en en en en en en en en en en en en en en en en en en en en en en en...

**Typical SCRUB Gibberish Example**

[Gibberish: High Fragmentation]

, Sin to,.-inA D R R. T A: D.A- B. HAa.,i '. W and M.1, H FA and.,,.. The,A S H...

**Typical NPO Gibberish Example**

[Gibberish: Repeated Sentences]

Being a civil engineer is a highly respected profession... Moreover. It instilled a sense of discipline. Moreover. It instilled a sense of discipline. Moreover. It instilled a sense of discipline.

**Typical DareU Gibberish Example**

[Gibberish: Repeated Phrases]

Some of the Geology books written by Hinah AmeEN include 'Stones, Stones Everywhere', 'Street Geology', 'Stuck in KarAChi Karst', 'Staircases in KarAChi Karst', 'Street KarAChi Karst Geology', and 'Still KarAChi Karst'.

*Figure 8.* Visualization of the gibberish responses from different unlearning approaches.

