# OpenReview forum: "De-attribute to Forget for LLM Unlearning"
_ICML.cc/2026/Conference — ICML 2026 regular_

### Official Review · Reviewer_XffK · 2026-02-27

**Soundness:** 4
**Presentation:** 3
**Significance:** 3
**Originality:** 3
**Overall Recommendation:** 4
**Confidence:** 5

**Summary:**

This paper introduces DareU, a reinforcement learning-based framework designed for large language model (LLM) unlearning. Rather than using the standard approach of maximizing prediction loss on the data intended to be forgotten—which often causes the model to output gibberish or suffer from overall performance degradation—the authors reframe the unlearning objective as a data de-attribution problem.

**Compliance With Llm Reviewing Policy:**

Affirmed.

**Final Justification:**

The authors' rebuttal effectively addressed my main concerns, particularly through the cross-evaluation experiment validating against reward hacking (W1), the C_seg sensitivity ablation (W2), and the additional clarifications on scalability and WASA metric behavior in the follow-up round. I maintain my positive recommendation with increased confidence.

**Key Questions For Authors:**

The framework relies on a lightweight classifier, $DA_c$, as an efficient approximation of an ideal attribution function. Is there a risk that the PPO algorithm is simply learning to "trick" this specific classifier rather than genuinely unlearning the data? Have you considered verifying the final unlearned model using a completely different, hold-out attribution metric to confirm the forgetting is genuine?

DareU takes longer than standard gradient ascent methods due to the autoregressive generation required by PPO. While it is definitely faster than full retraining, how do you see this scaling if a practitioner needs to unlearn millions of tokens? Do you think the de-attribution objective could be adapted for reward-free RL methods to cut down on compute?

In Appendix B, the log-based transformation uses a specific constant, $C_{seg} = 1.05$, to assign larger penalties to generations attributed to the forget set. How sensitive is the overall unlearning performance to this exact value?

**Limitations:**

The authors do briefly touch on the computational trade-offs and openly acknowledge that their classifier is an imperfect approximation. They also included a short Impact Statement regarding data privacy.

However, a dedicated and thorough limitations section is currently missing.

I recommend adding a specific "Limitations" paragraph that explicitly discusses the potential vulnerabilities of using a proxy reward model (e.g., reward hacking risks). It would also be helpful to openly discuss the hard boundaries of the method's scalability when dealing with massive forget sets where PPO's generation costs might become prohibitive.

**Strengths And Weaknesses:**

Soundness

Strength: The empirical setup is solid. Testing across different model architectures (Llama2, Qwen3) and diverse tasks (QA and completion) builds confidence in the results. Furthermore, the trick of sampling the attribution score every $\kappa$ tokens and broadcasting it forward is a very smart design choice to reduce the inference bottleneck during PPO. The ablation studies on sequential unlearning and semantic overlaps are also a nice touch.

Weakness: The entire framework heavily depends on the accuracy of the proxy classifier ($DA_c$). Because this classifier is trained on limited data, there is a risk that the PPO process might simply learn to exploit the classifier's blind spots (reward hacking) rather than genuinely unlearning the underlying concepts. While adversarial probing is briefly discussed, a deeper dive into the failure modes of this classifier approximation would strengthen the claims.

Presentation

Strength: The narrative flows well and is easy to follow. The authors do a good job of setting up the limitations of current methods and logically leading the reader to their proposed solution. The visual aids, particularly the breakdown of the PPO reward formulation, are helpful for understanding the mechanics.

Weakness: The discussion on how certain hyperparameters are selected could be a bit more transparent. For instance, a bit more intuition on why the specific log-based transformation ($C_{seg} = 1.05$) was chosen for the reward reshaping would be helpful for future reproducibility.

Significance

Strength: The problem domain is highly relevant right now, given the ongoing copyright and privacy debates surrounding LLM training data. Achieving unlearning without destroying the model's base knowledge is a known pain point, and this framework offers a workable, practical solution.

Weakness: The reliance on autoregressive generation during PPO makes the unlearning process computationally heavy compared to simpler, gradient-ascent-based methods. While the authors correctly note it is still faster than full retraining, scaling this framework to handle massive, real-world forget sets might be tough in practice.

Originality

Strength: The shift in perspective is quite refreshing. Moving away from raw loss manipulation or forced refusal generations (like training the model to just say "I don't know") and instead framing unlearning as minimizing attributability is a natural way to look at the problem.

---

> ### Author Rebuttal · Authors · 2026-03-31
>
> Thank you for recognizing our solid empirical setup, comprehensiveness, highly relevant problem domain, and refreshing perspective.
>
> ---
>
> > [W1] risk of reward hacking
> > [Q1] verify unlearning using different hold-out attribution metrics
>
> Thank you for the suggestions! We suspect that the risk may be lower when the classifier has high accuracy (e.g., ~80% accuracy in our experiments) and fewer blind spots.
>
> We first investigate whether it is possible to do reward hacking, i.e., reduce the attribution score without forgetting the underlying concepts. This can be checked by paraphrasing the generated responses semantically while preserving the underlying concepts. As shown below, such paraphrasing can reduce attribution scores, but it also produces less relevant content with a lowered ROUGE scores, i.e., forgetting occurs and the hack is not effective.
>
> ||Attr.|ROUGE
> -|-|-
> DareU|0.4160|0.3892
> DareU on paraphrased|0.3176|0.3056
>
> To address Q1, in the table below, we consider DareU using various DA classifiers (rows) and measure the attribution scores using a different hold-out metric based on another DA classifier (columns). Results show that our **DareU also effectively reduces attribution on hold-out metrics**. Thus, the PPO process has not inaccurately exploited the classifier's blindspots.
>
> ||LM Classifier metric|ST Classifier metric|WASA metric
> -|-|-|-
> Original|0.6576|0.8096|0.8390
> DareU LM Classifier|0.4160|0.4164|0.7229
> DareU ST Classifier|0.4074|0.4011|0.6727
> DareU WASA|0.4336|0.4525|0.7601
>
> ---
>
> > [W2] discussion on how certain hyperparameters are selected
> > [Q3] sensitivity to $C_{seg}$?
>
> We will add more discussion and ablation in our revision. The parameter $C_{seg}$ controls the boundary points in Fig 4 where the transformed DA score transitions from the non-constant logarithmic curve to either a constant 0 or -1. We should choose $C_{seg}$ such that the transformed DA score is not -1 from 0 to some boundary point between [0.5,1]. If the boundary point is lower (i.e., more -1), the reward signal would be flatter and less informative; if it is larger (i.e., less -1), the LLM may not be sufficiently discouraged from generating responses that are highly attributable to the forget set. In the table below, we evaluate DareU using different $C_{seg}$ and its corresponding values of clipped point. Results show that DareU is not sensitive to the choice of $C_{seg}$ in this range.
>
> $C_{seg}$|Boundary point|$D_f$ ROUGE|$D_r$ ROUGE|ToW
> -|-|-|-|-
> 1.05 (Ours)|0.65|0.407|0.797|0.816
> 0.7|0.5|0.413|0.788|0.815
> 1.61|0.8|0.411|0.802|0.832
>
> ---
>
> > [W3] scaling to unlearn millions of tokens
> > [Q2] adapt de-attribution objective for reward-free RL methods
>
> When forget sets are large and contain millions of tokens, we can adopt sequential unlearning (Sec 5.4/App D.4) to unlearn the full forget set in multiple rounds. In this way, in each unlearning round, fewer tokens are considered in $\mathcal{J}(\theta)$, allowing DaureU to scale effectively. As demonstrated in Fig 5, our DareU achieves effective unlearning performance close to that of Retraining in this setting.
>
> For Q2, it might be possible to design the de-attribution objective for reward-free RL methods, which usually rely on an external preference dataset. The design challenge would be how to translate the de-attribution objective into the external preference dataset effectively.
>
> ---
>
> > [L1] add a specific "Limitations" paragraph
>
> Thank you for your suggestions. In the revision, we will frame them as formal limitations and add the following:
> * DareU will be preferred over simpler loss-based approaches when (i) better unlearning performance that balances forget quality and model utility is desired and (ii) the users have higher computational budget and optionally the ability to exploit high-throughput inference engines to accelerate LLM generations. The user is free to choose from the trade-off based on their computation budget.
> * Future work should develop better and more efficient attribution functions. We will include more failure modes such as inaccurate classifiers.
> * We will add "The practitioner must have access to the pretraining or finetuning dataset in order to obtain a data attribution function or classifier. While this might exclude the application of DareU for ensuring successful pre-training data removal, DareU is still useful for many other real world applications such as when healthcare firms have to unlearned private health data that they finetune their LLM on."
> * The impact statement will contain potential negative consequences: "Users of DareU should be cautious that data de-attribution does not certify unlearning or data removal compliance and should instead verify unlearning through other methods. The attribution classifier should be protected to prevent adversarial uses such as membership inference."
>
> ---
>
> Thank you again for reviewing! We hope our clarifications and experiments can adress your concerns. We welcome further questions.

---

> > ### Author Rebuttal · Reviewer_XffK · 2026-04-02
> >
> > Thank you for your response. The cross-evaluation experiment for W1/Q1 is convincing and effectively addresses the reward hacking concern. The C_seg ablation (W2/Q3) also shows sufficient robustness.
> > However, the scalability argument (W3/Q2) remains weak — the sequential unlearning experiment (5 rounds × 80 samples) is far from the millions-of-tokens regime. I would also like the authors to comment on why de-attribution transfers less effectively to the WASA metric in the cross-evaluation matrix (Original 0.839 → 0.723 vs. much larger drops for LM/ST classifiers).

---

> > > ### Author Response · Authors · 2026-04-05
> > >
> > > Thank you for your reply. We are addressing your further questions below.
> > >
> > > ---
> > >
> > > > scalability: unlearning millions-of-tokens
> > >
> > > Thanks for the interesting question! To allow DareU to unlearn millions of forget tokens, a practitioner can consider the following possible solutions:
> > > * Perform sequential unlearning over 5 steps (or more), such that the number of tokens to be unlearned in each round of unlearning is 1/5 the total number of tokens. This makes the unlearning task easier to handle per round.
> > > * Instead of using the LLM-generated response $y_d$ for all forget data, we can use the attribution score of the original model's target response as a proxy as a mixture of SFT and RL. This implicitly assumes that a higher attribution score of the original model's target response correlates with a higher attribution score of the generated response. By directly using the target response, we can save the computation for autoregressive generation.
> > > * We can use the losses/attribution scores before unlearning to order and prioritize data to be unlearned.
> > >
> > > However, we believe that unlearning millions of forget tokens is unlikely in real applications due to the following reasons:
> > > * Unlearning requests are usually to remove a small portion of data that has privacy concerns or consists of incorrect/outdated/harmful information. These data are unlikely to be large in practice and unlearning them should not influence the original model usage. Unlearning a very large subset of data with millions of tokens is probably destroying the model, or can be as expensive as retraining the model.
> > > * On our arxiv dataset, the full dataset consists of around 2 million tokens, but the forget set, which is 5% of the full dataset, contains around 0.1 million tokens. This is in line with the scale of other popular LLM unlearning papers [1, 2, 3, 4].
> > >
> > > [1] Pratyush, Maini, et al. (2024). TOFU: A task of fictitious unlearning for
> > > LLMs. In Proc. COLM.
> > >
> > > [2] Ruiqi, Zhang, et al. (2024). Negative Preference Optimization: From Catastrophic Collapse to Effective Unlearning. In Proc. COLM.
> > >
> > > [3] Jinghan, Jia, et al. (2024) SOUL: Unlocking the Power of Second-Order Optimization for LLM Unlearning. In Proc. EMNLP.
> > >
> > > [4] Yuanshun, Yao, et al. (2024). Large language model unlearning. In Proc. NeurIPS.
> > >
> > > ---
> > >
> > > > why de-attribution transfers less effectively to the WASA metric
> > >
> > > This is because the WASA metric, when used as a measure of de-attribution, is less sensitive to the unlearning of the generated responses, likely due to their ability to learn an accurate mapping from texts to watermarks. In the following experiment, we analyze how the unlearning of the generated responses influences the different metrics. Specifically, we systematically approximate different levels of unlearning by masking k words from the original model's responses and measure the different metrics on the masked responses. The results below confirm that the WASA metric value reduces more slowly with the levels of unlearning, which explains why the WASA metric appears to be larger than other metrics in our previous experiments.
> > >
> > > masking k words|LM metric|ST metric|WASA metric
> > > -|-|-|-
> > > 1|0.6143|0.7740|0.8209
> > > 2|0.5774|0.7199|0.8099
> > > 3|0.5466|0.7054|0.7945
> > > 4|0.5241|0.6869|0.7802
> > > 5|0.4968|0.6463|0.7479
> > > 6|0.4802|0.6079|0.7345
> > > 7|0.4274|0.6118|0.7427
> > > 8|0.4341|0.5317|0.7070
> > > 9|0.4087|0.5045|0.6829
> > > 10|0.3475|0.4965|0.6478
> > >
> > > ---
> > >
> > > Thank you very much! We hope our additional explanations and experiments can address your further questions.

---

### Official Review · Reviewer_4fd2 · 2026-03-13

**Soundness:** 3
**Presentation:** 3
**Significance:** 3
**Originality:** 3
**Overall Recommendation:** 4
**Confidence:** 4

**Summary:**

This paper leverage data attribution as a guiding signal for model unlearning. Specifically, the authors propose DareU, a framework that incorporates attribution information to guide the unlearning process of PPO on fine-tuned models. The key idea is to shift the optimization target from directly modifying model outputs to reducing the attribution of the model to the forget data.

The paper evaluates the proposed method on two types of benchmarks, covering different forms of forget data, including query-based and text-completion-based settings. In addition to standard unlearning evaluations, the authors further assess the robustness and effectiveness of their approach from multiple perspectives, such as sequential unlearning scenarios and relearning attacks, and compare their method against existing baselines.

**Compliance With Llm Reviewing Policy:**

Affirmed.

**Final Justification:**

Thanks for author's response. I appreciate the clarifications provided, and I am satisfied with how my concerns were addressed. I will keep my positive score unchanged.

**Key Questions For Authors:**

See Strengths And Weaknesses* part.

**Limitations:**

yes

**Strengths And Weaknesses:**

**Soundness:**

**Strengths:**

	1. From a methodological perspective, using data attribution as the core idea is a smart design choice.

	2. The authors provide sufficient ablation studies for each component of the method, which helps clearly understand the role and contribution of each part.

	3. The authors explicitly acknowledge the time cost limitation of their approach. However, considering the experimental performance achieved, this trade-off is acceptable.


**Weaknesses:**

	1. Regarding the baseline results, for example the NPO method, is it possible that by tuning its hyperparameters it could also achieve performance close to the retrained model under the unlearning performance metric? In addition, I noticed that different methods in the appendix are trained with different numbers of epochs. Is there a specific reason for this experimental setting?

	2. One important motivation for avoiding over-forgetting is to prevent the model from producing repeated tokens or gibberish outputs after excessive unlearning. However, this aspect does not seem to be sufficiently discussed in the paper. It would be helpful to include qualitative examples as well as quantitative metrics to evaluate this phenomenon.

**Presentation:**

The paper is well written and has a clear organizational structure, making it easy to follow.

**Significance:**

This work focuses on the over-forgetting issue in the unlearning literature. Previous papers, even when using a retrained model as a baseline, often focus primarily on minimizing unlearning performance, while overlooking the importance of the retrained model as a reference point. From this perspective, the paper provides meaningful insights for the development of the field.

However, I have a major concern regarding the overall significance of the work. The paper imposes a strong limitation:
“We do not consider unlearning pre-trained LLMs in this work because pre-training employs public datasets that often raise fewer data concerns.” The discussion supporting this design choice is insufficient. It does not convincingly justify completely excluding pretrained model unlearning from the scope of the study. A large portion of prior work in this field focuses specifically on unlearning in pretrained LLMs, which remains an important and practical problem. If the authors could provide a more convincing justification or apply their method to pretrained models, I would be willing to reconsider and potentially increase my score.


**Originality:**

The paper proposes a well-designed approach and demonstrates strong performance on multiple benchmarks.

---

> ### Author Rebuttal · Authors · 2026-03-31
>
> Thank you for recognizing our core idea, sufficient ablation studies, and the trade-off between efficiency and effectiveness as strengths of our work.
>
> ---
>
> > [W1] tuning NPO hyperparameters to improve performance; reason for different numbers of epochs
>
> App. C.2 describes how we have tuned the hyperparameters for the baseline methods **within a controlled budget** (e.g., by observing the performance of using 4 learning rates). We have also tuned the number of epochs between 1-3 despite noting that fine-tuning was methods like Fine-tuning are designed for 1 epoch and others like GA severely degenerate with more epochs.
>
> We agree that more careful tuning can improve performance (e.g., NPO achieves 0.539 ToW at 1.3e-5 LR). However, exhaustive tuning is computationally expensive and often infeasible for real-world unlearning. Therefore, in our experiments, we standardized this budget to reflect practical constraints. In addition, DareU achieves a more stable performance across different hyperparameters than the baselines (App. D).
>
>
> ---
>
> > [W2] qualitative and quantitative evaluations of gibberish outputs
>
> Thanks for raising this insightful question! We will add a quantitative evaluation of whether the unlearned LLMs from our method and the baseline methods produce gibberish outputs. In the table below, we report the percentage of gibberish outputs generated across all forget queries. The table shows that **DareU produces less gibberish than most prediction loss-based unlearning methods**. Note that Finetune does not effectively unlearn $D_f$ and IDK forces 'I don't know' responses, hence producing 0% gibberish. Qualitative examples of the gibberish outputs are also shown in the [linked figure](https://ibb.co/23vZ9YRZ).
>
> ||ToW|gibberish (%)
> -|-|-
> Original|0.455|0
> Retraining|1.000|0
> Finetune|0.563|0
> GA|0.009|100
> GDiff|0.614|0
> SCRUB|0.219|45.25
> SCRN|0.251|10.75
> IDK|0.548|0
> NPO|0.239|23.58
> DareU|0.816|4.08
>
> Next, we check if DareU would produce more gibberish with smaller regularization ($\lambda_{dis}$) or more epochs (which has led to lower unlearning performance in Table 10). As shown below, $\lambda_{dis}$ barely affects the generation of gibberish, whereas training with more data epochs likely increase the gibberish \%. However, the \% remains lower than other methods and can be mitigated by using fewer epochs.
>
> $\lambda_{dis}$|gibberish (%)
> -|-
> 0.0|4.25
> 1.0|2.83
> 2.0(ours)|4.08
>
> data epoch|gibberish (%)
> -|-
> 1(ours)|4.08
> 2|6.08
> 3|8.42
>
> ---
>
> > [S1] a more convincing justification or apply DareU to pretrained models
>
> Thank you for raising this concern! Here we provide both further justifications and an attempt to apply our method to pretrained models.
>
> Justification on why focusing on finetuning on task-specific LLM is still significant:
> * In many real-life applications, users lack resources to train an LLM from scratch and finetune it on their task-specific dataset to adapt to their task at hand. For example, one may download a publicly available pretrained LLM (llama) and then finetune it on private healthcare data.
> * We focus on the downstream users who may need to ensure privacy compliance and unlearn part of the private fine-tuning data as in popular unlearning benchmarks and methods [1, 2].
> * Unlearning pre-training data involves larger datasets/token numbers and new non-trivial computation challenges that specific papers [3] address.
>
> We also agree that pretrained model unlearning is an important real-world application. In practice, DareU can be applied to unlearning pretrained models given both access to the pretraining dataset (or a subset of it) to train a data attribution classifier and sufficient compute for reinforcement learning. Initially, we excluded experiments on pretrained models as (i) they were excluded by some unlearning papers [4] and (ii) it is too expensive to obtain the *retrained* pretrained model for benchmarking/computing the ToW metric.
>
> Our previous experiments were based on the efficient LoRA finetuning instead. As a proof of concept for pretraining, we applied DareU to **full-parameter unlearning** of the TOFU dataset for Llama2-7B, where the model was initially full-parameter trained on the TOFU dataset. The table shows that DareU can still achieve high ToW score.
>
> Method|$D_f$ ROUGE|$D_r$ ROUGE|ToW
> -|-|-|-
> Original|0.932|0.926|0.379
> Retraining|0.341|0.994|1.000
> DareU|0.288|0.793|0.752
>
> ---
>
> We hope that the above justification and experiments address your concerns and improve your opinion of our work. We welcome further questions.
>
> ---
>
> [1] Pratyush, Maini, et al. (2024). TOFU: A task of fictitious unlearning for
> LLMs. In Proc. COLM.
>
> [2] Weijia, Shi, et al. (2025). MUSE: Machine Unlearning Six-Way Evaluation for Language Models. In Proc. ICLR.
>
> [3] Jin, Yao,et al. (2024). Machine unlearning of pre-trained large language models. In Proc. ACL.
>
> [4] Ruiqi, Zhang, et al. (2024). Negative Preference Optimization: From Catastrophic Collapse to Effective Unlearning. In Proc. COLM.

---

> > ### Author Rebuttal · Reviewer_4fd2 · 2026-03-31
> >
> > Thank you for your response. Most of my concerns have been addressed. I will raise my score.

---

### Official Review · Reviewer_a6RC · 2026-03-13

**Soundness:** 3
**Presentation:** 3
**Significance:** 3
**Originality:** 4
**Overall Recommendation:** 4
**Confidence:** 3

**Summary:**

This paper proposes a new unlearning methodology based on Data Attribution, which differs from existing unlearning approaches. Data Attribution represents how much a model's output is influenced by a particular dataset and is expressed as a scalar value between 0 and 1. A higher value indicates that the generated response is more likely to have been influenced by a specific dataset.
Existing unlearning approaches, particularly gradient-based methods, typically perform unlearning by increasing the loss on the forget set. However, such methods can achieve the objective even when the generated outputs collapse or when the model simply produces unrelated answers. In other words, the model may satisfy the optimization objective without actually removing the influence of the forget data.
To address this issue, the paper reformulates the objective of unlearning as reducing the Data Attribution score. Since the goal of unlearning is to make the model behave similarly to a model trained without the forget set, the authors argue that minimizing attribution to the forget dataset (i.e., de-attribution) better aligns with the true objective of unlearning.
To optimize this objective, the authors propose a framework called DareU, which uses a reinforcement learning approach based on Proximal Policy Optimization (PPO). In this framework, attribution scores are used as reward signals to guide the model to generate responses with lower attribution to the forget set. At the same time, the framework includes a distillation-based regularization term to preserve the model’s utility on the retain set.
The proposed method is evaluated on the TOFU and ArXiv benchmarks using LLaMA2-7B and Qwen3-8B models. The experiments compare DareU with several baseline unlearning methods including gradient-based approaches, second-order optimization methods, and preference-based approaches. The results suggest that DareU achieves a better balance between effective forgetting and preserving model utility.

**Compliance With Llm Reviewing Policy:**

Affirmed.

**Key Questions For Authors:**

1. Is de-attribution equivalent to unlearning?

The core idea of the paper assumes that reducing attribution scores corresponds to removing the influence of the forget dataset. However, it is not entirely clear whether de-attribution reliably approximates retraining-based unlearning. Could the authors provide additional evidence or analysis showing that reducing attribution correlates with the behavior of a retrained model?

2. Attribution reliability under similar datasets

How does the attribution classifier behave when different data owners contain highly similar or overlapping datasets? If two owners contain similar content, it is possible that attribution scores become ambiguous. Additional analysis in such settings would help clarify whether attribution reliably reflects the removal of the forget dataset’s influence.

3. Robustness against generation collapse

The paper points out that gradient-based unlearning methods may lead to generation collapse while still satisfying the objective. Could the proposed PPO-based framework exhibit similar behavior when optimizing attribution-based rewards? It would be useful to analyze whether the model sometimes learns degenerate generation patterns that simply reduce attribution scores.

**Limitations:**

1. Computational cost and scalability concerns.

DareU incurs significantly higher computational costs than prediction loss-based approaches due to token-by-token autoregressive generation required by PPO (as shown in Table 4, DareU is substantially slower than GA, NPO, and most baselines). The authors acknowledge this but do not discuss it as a formal limitation. For longer sequences (e.g., the ArXiv dataset), the cost becomes particularly prohibitive, raising questions about the practicality of DareU for real-world unlearning scenarios involving large-scale datasets or production LLMs.

2. Reliance on an imperfect attribution classifier.

The entire framework hinges on DA_c, a lightweight LLM classifier that serves as an approximation of an ideal attribution function. The authors acknowledge this approximation may be imperfect, yet they do not discuss potential failure modes — for instance, when forget and retain data share significant semantic overlap, the classifier may systematically misattribute responses. The robustness test in App. D.5 is limited to one synthetic setting, and the risk that the LLM could learn to game the classifier (producing low attribution scores without genuinely unlearning) deserves deeper discussion.

3. Narrow evaluation scope and missing societal impact discussion.

Experiments are limited to two benchmarks (TOFU and ArXiv) with relatively small forget sets, both involving fine-tuned rather than pre-trained LLMs. The authors explicitly exclude pre-training unlearning from scope but do not discuss how this limits the applicability of DareU to real-world privacy compliance scenarios (e.g., GDPR requests on foundation models). Furthermore, the Impact Statement does not address potential negative consequences, such as the risk of providing a false sense of data removal compliance or the possibility that adversaries could exploit the attribution framework to identify which data was used for training.

**Strengths And Weaknesses:**

# Strength
1. Novel perspective on the unlearning objective

The paper introduces a new perspective on the objective of machine unlearning by framing it as a data de-attribution problem. This perspective is both original and conceptually interesting. Instead of maximizing loss on the forget set, the paper proposes directly minimizing attribution to the forget dataset. This reframing attempts to align the optimization objective with the actual goal of unlearning: removing the influence of specific training data while maintaining general model capability.

2. Balanced treatment of forgetting and model utility

The proposed framework explicitly considers the trade-off between effective forgetting and preserving model utility. The method combines PPO-based optimization that minimizes attribution to the forget set with distillation-based regularization that preserves the behavior of the original model on the retain set. This design reflects a practical understanding of the unlearning problem.

3. Comprehensive empirical evaluation

The paper evaluates the proposed approach against multiple representative baselines including gradient-based methods, second-order methods, and preference-based approaches. Several evaluation metrics such as ROUGE alignment, Truth Ratio, Extraction Strength, and membership inference attacks are used to assess both forgetting quality and utility preservation.

4. Practical implementation of attribution estimation

Since exact attribution functions are computationally infeasible, the paper proposes a practical approximation using a lightweight classifier trained with LoRA. This allows the framework to be implemented at scale while acknowledging real-world constraints.

# Weakness
1. Reliability of attribution as a proxy for true data influence

A central question that remains unclear is whether de-attribution is truly equivalent to unlearning. While the proposed objective minimizes the attribution score of the forget dataset, it is not fully evident that reducing attribution necessarily corresponds to removing the actual influence of the forget data from the model.
Additionally, if different data owners contain similar or overlapping datasets, the attribution classifier may produce similar scores across those owners. In such cases, reducing attribution for a specific dataset may not guarantee that the actual influence of the forget dataset has been removed. Evaluating the behavior of the attribution function under datasets with overlapping distributions would strengthen the paper.

2. Potential generation collapse similar to gradient-based methods

As pointed out in the paper, gradient-based unlearning methods may achieve the optimization objective even when generation collapses or when the model produces irrelevant answers. Although the proposed approach uses attribution-based rewards instead of loss maximization, similar collapse behavior could still occur. Since the model optimizes a scalar reward signal derived from attribution scores, the model may learn degenerate generation strategies that reduce attribution without actually removing knowledge related to the forget set.

---

> ### Author Rebuttal · Authors · 2026-03-31
>
> Thank you for the detailed review that recognizes the novelty and comprehensiveness of our work.
>
> ---
>
> > [W1] reliability of attribution as a proxy for true data influence
> > [Q1] Is de-attribution equivalent to unlearning?
>
> Like increasing the forget loss or decreasing the retain loss, de-attribution is also not equivalent to true unlearning, i.e., perfectly approximating a retrained LLM. Our response will focus on justifying why it can be a reliable proxy.
> * In Tables 2, 3 and 14, we empirically observe that lower attribution scores and DareU lead to closer performance (e.g., ROUGE) to retrained LLM and better ToW.
> * Currently, there is no standard metric to measure true unlearning and no benchmark datasets or models with different levels of true unlearning. Thus, to verify if de-attribution *correlates* with true unlearning, we systematically construct different levels of true unlearning by masking the forget responses. Specifically, we mask k words from the forget responses and measure attribution score and ROUGE on the masked responses. From the [linked figure](https://ibb.co/PGg76VMs), **as the attribution score decreases, the ROUGE score (which measures similarity of content and semantics) decreases as well**, suggesting that content is also unlearned.
> * The table below comparing different unlearning methods also reflects that **lower attribution scores are correlated with lower $D_f$ ROUGE**.
>
> ||Attr. Score|$D_f$ ROUGE
> -|-|-
> Original|0.6576|0.908
> Fine-tune|0.6338|0.772
> GA|0.0014|0.000
> GDiff|0.4250|0.419
> SCRUB|0.6389|0.268
> SCRN|0.6401|0.292
> IDK|0.2347|0.028
> NPO|0.0922|0.286
> DareU|0.4160|0.407
>
> We believe it is important to note that **other loss functions also do not reliably imply true unlearning in all cases**. Sec. 3 describes how GA may prefer collapsed LLMs over retrained LLMs. Additionally, the retain loss used by fine-tuning does not consider if the LLM has sufficiently unlerned the forget data, resulting in a high $D_f$ ROUGE in Tables 2 and 3.
>
> ---
>
> > [W1] [Q2] attribution reliability under similar datasets
>
> Thank you for your insightful suggestion and we will discuss this explicitly in our revision. We agree that overlapping distributions may result in lower accuracy for the DA classifier and make unlearning harder. Thus, we have considered semantic overlapping distributions in App D.5, Table 14. The overlap causes the ToW of DareU to slightly drop from 0.816 to 0.797, and the classifier accuracy to drop from 0.877 to 0.723. However, **DareU still has good unlearning performance --- it outperforms almost all baseline methods and achieves the second highest ToW score out of all methods.** Moreover, the overlapping distribution in Table 14 hurts the unlearning performance of almost all baseline methods (compare Table 14 vs Table 2). Thus, the limitation is not unique to DareU.
>
> ---
>
> > [W2] [Q3] robustness against generation collapse
>
> Thank you for raising this interesting question! We add an experiment for quantitative evaluation of whether the unlearned LLMs from both our method and the baseline methods produce gibberish outputs. In the table below, we report the percentage of gibberish outputs generated across all forget queries. Qualitative examples of the gibberish are shown in the [linked figure](https://ibb.co/23vZ9YRZ). The results show that **DareU produces less gibberish than most prediction loss-based unlearning methods**. This may be due to our design of the distillation regularization term at the end of page 5. Note that Fine-tune does not effectively unlearn $D_f$, and IDK makes the unlearned model outputs "I don't know", hence producing 0 percent gibberish in the generated outputs.
>
> ||ToW|pct. gibberish
> -|-|-
> Original|0.455|0
> Retraining|1.000|0
> Fine-tune|0.563|0
> GA|0.009|100
> GDiff|0.614|0
> SCRUB|0.219|45.25
> SCRN|0.251|10.75
> IDK|0.548|0
> NPO|0.239|23.58
> DareU|0.816|4.08
>
> In addition, we check if it is possible for DareU to produce more generation collapse by shrinking the regularization term or training for more iterations, which has led to lower unlearning performance in Table 10. The table below shows that the regularization term does not affect the % with generation collapse, while training with more data epochs may increase it. However, the % is lower than other methods and can be mitigated by using fewer epochs.
>
>
> $\lambda_{dis}$|generation collapse (%)
> -|-
> 0.0|4.25
> 1.0|2.83
> 2.0(ours)|4.08
>
> data epoch|generation collapse (%)
> -|-
> 1(ours)|4.08
> 2|6.08
> 3|8.42
>
> ---
>
> > discussions of the limitations:
>
> Thank you for acknoweledging that we have raised limitations 1 and 2 in the paper. In the revision, we will frame them as formal limitations into a dedicated section. Please refer to our response to Revewer XffK for detailed text on these formalized limitations.
>
> ---
>
> Thank you for your reviews. We hope that our clarifications and additional empirical results have addressed your concerns. We would be happy to provide further clarification.

---

### Official Review · Reviewer_WXHV · 2026-03-13

**Soundness:** 3
**Presentation:** 4
**Significance:** 3
**Originality:** 3
**Overall Recommendation:** 4
**Confidence:** 2

**Summary:**

This paper proposes DareU, a reinforcement learning framework for LLM unlearning that replaces traditional loss-based objectives with a data de-attribution objective. The method uses an attribution model to provide reward signals within a PPO-based optimization, with an additional distillation term to preserve utility on the retain set. Experiments suggest that DareU can achieve a favorable balance between forgetting quality and model utility compared to several baselines.

**Compliance With Llm Reviewing Policy:**

Affirmed.

**Final Justification:**

The rebuttal addressed my main concerns with useful clarifications, especially on the relation between de-attribution and unlearning. The responses improved my understanding of the method. I keep my score: the paper presents a novel and technically solid idea with promising results, though it still has some limitations in practicality and scope.

**Key Questions For Authors:**

1.	Can the authors clarify under what conditions reducing attribution is expected to faithfully approximate retraining-based unlearning?
2.	How sensitive is DareU to inaccuracies in the attribution classifier?
3.	In what practical settings would DareU be preferred over simpler loss-based approaches given the added cost?

**Limitations:**

Yes

**Strengths And Weaknesses:**

**Strengths:**
- The idea of framing unlearning as reducing attribution rather than maximizing loss on the forget set is conceptually appealing.
- The overall pipeline (attribution model, PPO optimization, and distillation regularization) is clearly described.
- The experimental evaluation covers multiple datasets, models, and baselines.

**Weaknesses:**
- As a reader not deeply specialized in attribution methods, I am not fully convinced that lowering attribution scores reliably implies true unlearning in all cases.
- The approach depends heavily on the learned attribution classifier, and the robustness to attribution errors is not thoroughly analyzed.
- The PPO-based solution introduces noticeable computational overhead and the practical advantage over simpler methods could be better justified.

---

> ### Author Rebuttal · Authors · 2026-03-31
>
> Thank you for your helpful review of our work and appreciating the novelty, clarity, and extensive experimental evaluation of our work! We will address your questions below.
>
> ---
>
> > [W1] As a reader not deeply specialized in attribution methods, I am not fully convinced that lowering attribution scores reliably implies true unlearning in all cases.
> > [Q1] Can the authors clarify under what conditions reducing attribution is expected to faithfully approximate retraining-based unlearning?
>
> We understand your concern that lowering attribution scores may not *always* imply "true unlearning".
> * Lowering attribution is expected to do so and better approximate the retrained LLM under the condition that **the attribution classifier accurately assigns high scores to responses influenced by the forget dataset and low scores to responses that are not (e.g., responses to queries about the retain set by the retrained LLM)**. For example, the classifier would be more accurate when the forget and retain dataset content significantly differ.
> * In Tables 2, 3 and 14, we also empirically observe that lower attribution scores and DareU lead to closer performance (e.g., ROUGE) to retrained LLM and better ToW.
> * Currently, there is no standard metric to measure true unlearning and no benchmark datasets or models with different levels of true unlearning. Thus, to verify if de-attribution *correlates* with true unlearning, we systematically construct different levels of true unlearning by masking the forget responses. Specifically, we mask k words from the forget responses and measure attribution score and ROUGE on the masked responses. From the [linked figure](https://ibb.co/PGg76VMs), **as the attribution score decreases, the ROUGE score (which measures similarity of content and semantics) decreases as well**, suggesting that content is also unlearned.
> * The table below comparing different unlearning methods also reflects that **lower attribution scores are correlated with lower $D_f$ ROUGE**.
>
> ||Attr. Score|$D_f$ ROUGE
> -|-|-
> Original|0.6576|0.908
> Fine-tune|0.6338|0.772
> GA|0.0014|0.000
> GDiff|0.4250|0.419
> SCRUB|0.6389|0.268
> SCRN|0.6401|0.292
> IDK|0.2347|0.028
> NPO|0.0922|0.286
> DareU|0.4160|0.407
>
> Lastly, we believe it is important to note that **other loss functions also do not reliably imply true unlearning in all cases**. Sec. 3 describes how GA may prefer collapsed LLMs over retrained LLMs. Additionally, the retain loss used by fine-tuning does not consider if the LLM has sufficiently unlearned the forget data, resulting in a high $D_f$ ROUGE in Tables 2 and 3.
>
> ---
>
> > [W2] The approach depends heavily on the learned attribution classifier, and the robustness to attribution errors is not thoroughly analyzed.
> > [Q2] How sensitive is DareU to inaccuracies in the attribution classifier?
>
> Thank you for raising this concern! We have considered different attribution classifiers in Table 5 and they generally have good accuracy (~80%) and consistently lead to better performance than other unlearning methods (Table 6 vs Table 2). WASA has the lowest accuracy but still a relatively high ToW.
> We will include a new experiment where we add noise to the classifier output/scale the temperature and artifically reduce the accuracy and measure the unlearning performance. As shown in the table below, the ToW score tends to drop with lower accuracy but **DareU still achieves a relatively high ToW across classifiers with different accuracies**. This suggests that DareU is not very sensitive to inaccuracies in the attribution classifier.
>
> Classifier Acc.|$D_f$ ROUGE|$D_r$ ROUGE|ToW
> -|-|-|-
> 0.877 (Ours)|0.407|0.797|0.816
> 0.814|0.413|0.786|0.802
> 0.760|0.425|0.791|0.800
> 0.678|0.413|0.783|0.795
> 0.488|0.440|0.769|0.771
>
> ---
>
> > [W3] The PPO-based solution introduces noticeable computational overhead and the practical advantage over simpler methods could be better justified.
> > [Q3] In what practical settings would DareU be preferred over simpler loss-based approaches given the added cost?
>
> Thank you for raising this concern. We attempted to justify the higher computational overhead at the end of Sec 5.2. The key idea is that **higher cost leads to better performance** and the **user can choose their desired trade-off**, which Reviewer 4fd2 also considers acceptable. DareU will be preferred over simpler loss-based approaches when (i) better unlearning performance that balances forget quality and model utility is desired and (ii) the users have a higher computational budget and optionally the ability to exploit high-throughput inference engines to accelerate LLM generations.
>
> ---
>
> Thank you again for reviewing our work. We hope that the above justifications and experiments have addressed your concerns. We welcome any further questions.

---

> > ### Author Rebuttal · Reviewer_WXHV · 2026-04-03
> >
> > Thank you for the clear and detailed responses to my questions. Most of my concerns are addressed.

---

### Decision · Program_Chairs · 2026-04-30

**Decision:**

Accept (regular)

**Comment:**

This paper proposes an RL-based LLM unlearning framework that replaces traditional unlearning objectives with a data de-attribution objective. On the positive side, reviewers agree that the method design is smart and novel, the empirical evaluation is solid and comprehensive, and the paper is clearly written. Prior to the rebuttal, reviewers also raised several concerns regarding the rationale of the new objective for unlearning, computational efficiency, and dependence on the proxy classifier. In the rebuttal phase, the authors provided careful analysis, incorporated additional experiments, and also responded carefully to follow-up reviewer concerns. As a result, all reviewers remain positive, with reviewer XffK increases confidence to 5. Given the consensus among all reviewers, the AC recommends accept.